# Construction Techniques and Detailing for Romanian *Paiantă* Houses: An Engineering Perspective

Andreea Dutu [1,*], Mihai Niste [1], Iolanda-Gabriela Craifaleanu [1,2] and Marina Gingirof [3]

1   Faculty of Civil, Industrial and Agricultural Buildings, Technical University of Civil Engineering Bucharest (UTCB), 020396 Bucharest, Romania
2   National Research and Development Institute in Construction, Urban Planning and Sustainable Spatial Development, URBAN-INCERC, 021652 Bucharest, Romania
3   Rural Working Group of The Order of Architects in Romania (OAR), 010312 Bucharest, Romania
*   Correspondence: andreea.dutu@utcb.ro

**Abstract:** Traditional houses represent landmarks of local cultures all over the world. In seismically prone countries, the traditional timber frames with different types of infills have shown quite good earthquake resilience, an essential feature considering their large number, their simplicity and their reduced cost, owing to the use of local materials and workmanship. In fact, their seismic behavior fostered the interest in further scientific research, including that addressing engineering aspects. Because of their diversity in layout and detailing, noticeable even among houses in the same area, developing general methods to preserve and strengthen such buildings is still a difficult task. This paper presents an overview of the traditional building construction techniques in Romania, focusing on the structural configuration and detailing of the so-called *paiantă* houses. Largely used all over the country, these houses have shown that they can generally withstand earthquakes at least without collapsing and, most frequently, with minor or repairable damage. Their preservation is nowadays a major challenge, as they are being gradually either demolished and replaced, or retrofitted by using invasive techniques. Their cultural and heritage value is undeniable, and there are still many things to be learned from the past craftsmanship and re-valuated in the future. Three case studies of rural buildings located in different areas of the country were chosen among the most widespread *paiantă* versions, and comprehensive engineering assessments were conducted to identify their detailing and degradations. Based on the most common degradations, the conclusions drawn from this study can be used to substantiate further research aimed at selecting the most appropriate construction and strengthening techniques.

**Keywords:** traditional; construction methods; timber frame; masonry; earth; degradations

## 1. Introduction

Traditional structures with timber frames and masonry or other types of infills are part of the national heritage of many countries. Even though their structural system is not specifically designed to withstand earthquakes, but only gravitational loads, it was shown that it can contribute to seismic performance as well [1]. Besides their intrinsic resistance, the gradual improvement over the years, or even centuries, of the conception of these houses, following the experience of earthquake damage, has played an important role. The seismic performance of this system was observed in strong earthquakes such as those of Kocaeli (1999), Kashmir (2005) and Haiti (2010). In the Izmit earthquake, even though heavily damaged, these types of houses did not collapse [2]. The good seismic performance of timbered masonry was also reported by Langenbach (2011) [3]. It is worth noting as well that timber framed masonry (TFM) has been used nowadays for the reconstruction of areas affected by strong earthquakes, in countries like Portugal and Pakistan [2].

During the past decade, several studies were conducted on vernacular practices in the field, with a special focus on the experimental assessment of their behavior. These

studies comprised the Haitian *kay peyi*, discussed in [4], the *casa baraccata* in Calabria, Italy, in [5] or the *dhajji dewari* in the mountainous areas of South Asia in [6,7]. A reference guidebook on *dhajji dewari*. aimed for the use of technicians and artisans, was prepared in [8], based on results from extensive studies including shake table tests and synthesizing the expertise gained from the over 120,000 rural houses that were rebuilt after the 2005 Kashmir earthquake using this construction technique. In [9,10] experimental research was performed on the traditional *Chuan Dou* timber structures; the Peruvian *quincha* buildings were studied in [11]; [12–14], among others, focused on Portuguese *pombalino* buildings or similar, while [15] shows the experimental research conducted on the Turkish *himis* buildings. Similar types of structural systems were used for centuries, under various names, also in other parts of the globe, as pointed out by Gülkan and Langenbach in [2]. A thorough field investigation on traditional houses in Greece was described in [16].

Timber frames with various infills represent an important part of the traditional housing stock in Romania. Even though based on different construction techniques, the houses of this type are known under the generic term of "paiantă". Several studies revealed that houses with timber frame and earth infill walls ("*paianta*") were built in Romania ever since the end of the early Neolithic [17]. Similar structures were identified in archeological sites dating from the Bronze Age and the Middle Age, most frequently located in the hilly and mountainous areas of the country. Comprehensive descriptions of various types of traditional buildings in Romania, viewed from engineering and architectural perspectives, can be found in [18–24]. At present, a relatively small number of publications in English language can be found on this subject, especially when seeking for an engineering perspective about their configuration and behavior under various actions.

According to data from the Romanian census of 2011—the most recent one for which complete data is available, 43% of the residential building stock is represented by traditional houses. Out of these, a percentage of 23% consists of timber-only structures, while the majority of 77% consists of timber frames with infills and adobe houses [25]. These houses were built, in general, during the post-WWII (1946–1960) period [26]. After 1960, in conjunction with the growing urbanization and industrialization of the country, the traditional methods were gradually abandoned in favor of masonry and reinforced concrete construction. At present, less and less country people still master the craftsmanship of properly building a traditional house. Consequently, the number of such buildings is rapidly decreasing, with owners demolishing or retrofitting them by invasive techniques.

Various initiatives were taken during recent years for saving traditional buildings in rural areas of the country, including TFM houses, and for their re-valuation as landmarks of Romania's cultural identity. From the perspective of today's exigences regarding building safety, comfort and cost, TFM houses are relatively cheap and easy to build, combining also the advantages of an ecologic approach, of a particular aesthetics and, most important for occupants' safety in earthquake-prone areas, of a satisfactory to good seismic performance, if properly detailed and maintained. The traditional building methods often include techniques that are transmitted, over time, from one generation to another and which have gradually established a local seismic culture [27]. Even though the reasons for applying certain techniques were lost many generations ago, rural builders still use them. This perishable resource should be re-evaluated from an engineering point of view and perpetuated within a novel approach combining old and new techniques to preserve and promote Romanian vernacular architecture.

One of the most recent research projects implemented in this regard was TFMRO, conducted at the Technical University of Civil Engineering Bucharest (UTCB) [28]. The experimental tests performed on timber frames with different infill types ("*paiantă*") demonstrated the good deformation capacity for such buildings. This was the first time such an engineering research program was focused on traditional Romanian buildings. Further research in this field, planned for the following years, will bring additional insights into their capability to meet nowadays housing requirements.

The paper presents the characteristic features of *paianta* houses, as observed during several field investigations. The field investigations were made with reference to the templates suggested in [29,30]. Due to the variation of the construction details even among the same area or the same typology, the collaboration between both engineers and architects was necessary to draw valid conclusions, use the same terms and understand the main details that classify a house into the *paianta* typology. To deepen the knowledge, three case studies were selected among the most common types of *paianta* and structurally assessed to observe the damage and degradations. These were analyzed comparatively and integrated into a list of possible damages specific to *paianta* houses, providing thus a comprehensive outline of the main aspects to focus on when investigating an existing house. This list can represent a useful tool for field investigations. Based on the degradation inventory, further research will be conducted to establish solutions for the most common problems occurring in these buildings, by using contemporary scientific knowledge and techniques.

## 2. Seismic Performance of Traditional Romanian Houses

In Romania, after the two major earthquakes that hit the country on 10 November 1940 and 4 March 1977, quite few cases of collapse or major damage were reported in traditional buildings with timber frame and masonry infill or other infills. On the contrary, special reference is made to the good seismic performance of some particular types of traditional houses in Romania in [31,32]. A relevant example, the case of the over 100 old traditional buildings at the Village Museum in Bucharest, which underwent only insignificant damage from the M = 7.2 March 4, 1977 earthquake, is mentioned in support of similar observations performed in other regions of the country. A possible explanation proposed by the above authors is that single family rural houses in Romania generally have a rectangular or regular plan layout and internal walls creating a "honeycomb" effect, which has favorable effects under seismic loads. However, the diversity of materials and construction methods used in Romania makes it difficult to categorize the response of rural houses to earthquakes. For instance, the cited works mention buildings with timber frame and earth infills ("*paianta*"), as well as wooden buildings or stone and wood buildings, as having behaved well in earthquakes. In contrast, for buildings made of adobe bricks ("*chirpici*"), rammed earth or cob, major damage and even collapse were reported. In a study on the vulnerability of various types of buildings, conducted after the 1977 earthquake, ref. [33] classifies timber frame and earth infills, together with adobe houses, in the category of buildings made of low-quality materials, most of which built before 1900. The authors note that earthquake damage observed in these buildings consisted of "a wide vertical crack at the intersection of walls, with a tendency to lose stability, and expulsion, inclined cracks, tendencies of sliding of floors and roofs, collapse of chimneys etc.". It should be mentioned, however, that this description of damage refers here to the entire category, and not specifically to timber frame and earth infills houses.

The spectral content of the ground motions affecting the area should also be considered in the analysis of the past seismic performance of these buildings, given that such types of low-rise buildings are particularly sensitive to high-frequency ground motion content. In case such spectral content was less represented in a certain area, the lack of significant damage could be also explained by this fact. This aspect was highlighted by [34], but also in other works on the seismic behavior of the existing building stock in Romania. Moreover, according to the Romanian seismic design code, P100-1/2013 [35], large areas in the Bucharest surroundings and in the southern and south-eastern part of Romania are classified as being prone to ground motions characterized by low-frequency components, thus less damaging for low-rise buildings. This factor should be however considered only in conjunction with other factors mentioned below, as these could prevail in certain cases.

One of the most important factors pointed out in the literature [36,37] is that certain traditional construction practices often provide resistance both to gravity and lateral loads, as well as good deformation capacity. The quality of materials and the overall layout and detailing of the structures also have an important contribution [15]. In [15] the main types of

traditional houses are compared in a representative rural area of Romania (Valea Doftanei) with the structural systems used in the same area nowadays. The description of old non-engineered buildings was based on earlier works [31,36]. The cited authors mention some features that led to the good seismic performance observed for these buildings, such as the relatively small plan dimensions and height (one story, rarely two storeys), the relative symmetry of the building layout, the light roofs, the wooden ring beams placed at the foundation and roof levels and the wooden horizontal elements at the corners of the walls. They also describe typical general damage observed in the analyzed types of buildings, among which roof and wall dislodging, stress concentration and cracks at the corners of the openings, including separate mentions about specific damage recorded following the more recent strong Vrancea (Romania) earthquakes (1977, 1986, 1990), such as chimney damage, roof displacement and plaster failure. A special note is made about the good performance of timber frame structures with infills or timber plated, which behaved better than masonry houses, as they generally withstood the earthquakes without collapse or major damage.

## 3. Traditional Houses: Components and Materials

Some specific components and building techniques of Romanian traditional houses are briefly presented in the following.

### 3.1. The Mudsill

According to [18] the mudsill (present only in post-1944 houses) was made of round or carved timber, from the possible following species: fir, beech, acacia, willow, oak, alder, hornbeam, ash, birch etc. (whatever was available). Usually, hardwood was used. The most common timber species for the mudsills were oak and holm. Mudsills could have been placed in different ways:

— directly on the ground in some areas (Argeş County, Dâmboviţa County, Buzău County etc.), especially for the ones with walls made of wattle and daub.
— on pilasters (called "*chei*" or "*lespezi*") of 40–50 cm, made of stone, clay/stone brick masonry or concrete (since 1930), placed only at the corners of the house. The stones can be placed sometimes also at the midway of the mudsill; after the walls are erected, the space between the big stones is filled with other stones (dry, unbinded), so that the construction props can be removed.
— on small piles (Figure 1) (called "*pari, gâşte, butuci, bulumaci, căsuţe, chituci, tufani*", depending on the area) were also used to support the mudsill and the post would be connected to the mudsill through a mortise ("*scob*"). The props are inserted into the soil about 90 cm deep and connected to the mudsill with 20 cm-long nails.
— on the foundation, to avoid capillary action (after 1916; in general, post-WWI). The foundation could be made either with river or quarry stone, concrete or burned clay brick masonry.

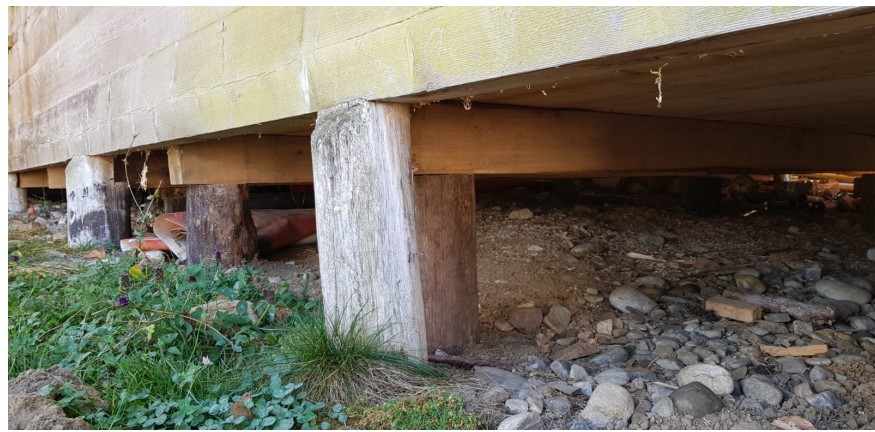

**Figure 1.** Timber piles foundation.

### 3.2. The Foundation

The foundation is usually present in houses built after 1944. Sometimes the posts were directly buried into the ground or a trench was dug and then filled with a mortar, called "*ciamur*", made of yellow soil (clay) mixed with straw or chaff (called "*pleavă*"). Where the groundwater level was high (less than 50 cm below ground surface), no trench was dug [18]. In some areas (i.e., Prahova County), a 50 cm-deep trench was dug and filled with gravel and the foundation was raised above the ground with about 50–150 cm. A slaked lime was poured on the gravel in the trench, whenever possible. Then, stones were cut to complete the foundation. The trench was dug usually until reaching the soil having a proper load bearing capacity (bedrock or loam), i.e., between 50 and 80 cm. Sometimes the trench was filled with rammed earth and had a width of 80 cm. Between the foundation trenches, rammed earth was placed. Some houses, in particular those of wealthier people, had cellars underneath some of the rooms.

The foundations, depending on the region of the country, could have been made of:

- river, quarry or carved stones;
- burned or unburned clay brick masonry;
- timber piles;
- oak timber combined with layers of quarry stone;
- concrete with large rocks (with coarse aggregates);
- adobe;
- grassy soil
- concrete or concrete ballast.

Sometimes the foundations were "dry", i.e., having no binder; this is the case when they were made of river stone. However, if a binder existed, it could have been made of:

- earth (soil) mixed with dung, straw or wheat chaff ("*pleavă*");
- mortar made of lime, sand or earth, sometimes with gravel (lime:sand proportion 1:3), sometimes with cement.

The exterior finishing of the foundation, if present, was made of earth ("*pământ*") mixed with cow dung and sand, sometimes with clay ("*lut*") and soot ("*funingine*").

### 3.3. The Walls

The walls of the traditional houses can be made of different structural systems. The structural systems can be as follows:

- earth;
- adobe bricks (Figure 2);
- stone walls (Figure 3);
- horizontal log walls ("*bârne*") (Figure 4);
- rammed earth, with yellow clay; in the Dobrogea area (southeastern Romania), a layer of horizontal reed ("*trestie*") or twigs was placed in the wall every 10 cm of rammed earth;
- clay mixed with straw, chaff ("*pleavă*"), manure ("*balegă*"), thistles ("*mărăcini*");
- alternated layers of stone and burned clay brick walls (Figure 5);
- wattle and daub ("*împletitură*" or "*grădele*"); it may contain diagonal braces (called "*paiante*") (Figure 6);
- vertical log walls ("*bârne*");
- timber braced "*paiantă*" walls with infills made of wattle and daub, horizontal or vertical planks, earth and straw, oven burned mud brick (Figure 7); unburned mud brick, horizontal timber strips on both sides of the posts infilled with mud and straw (called "*șipcuială*") (Figure 8);
- burned or unburned mud brick masonry walls.

The timber species used for the walls were most often oak and holm, but fir and spruce, or a mix of coniferous wood can also be used [20]. The timber was prepared much earlier than the start of the construction process. The craftsmen knew, from generation to generation, how to identify the good timber, when to cut it and how to dry it. In addition,

the trees on the top of the hills, the most exposed to wind, growing slower than the other, had the densest fiber. The timber was cut in the winter, when the trees have less sap. If the cut is made in the summer, the timber cracks significantly and this later facilitates the occurrence of biological decay [20]. The bark should be removed soon after cutting, so that the wood would dry faster and not rot. Figure 9 shows a good quality wood tree (left) and a bad quality one, showing rotten areas (right).

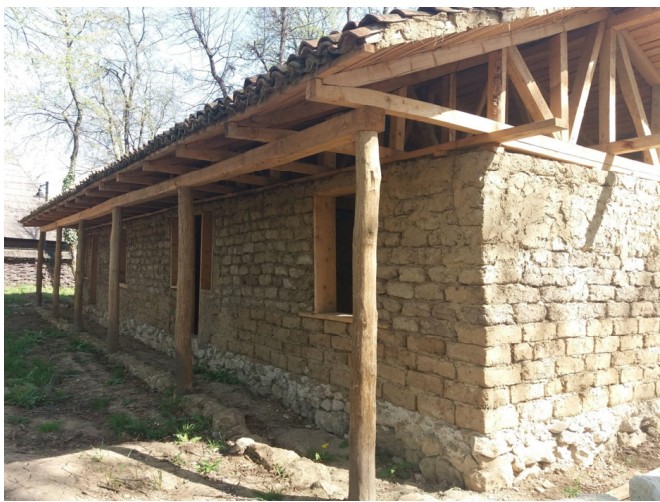

**Figure 2.** Adobe bricks.

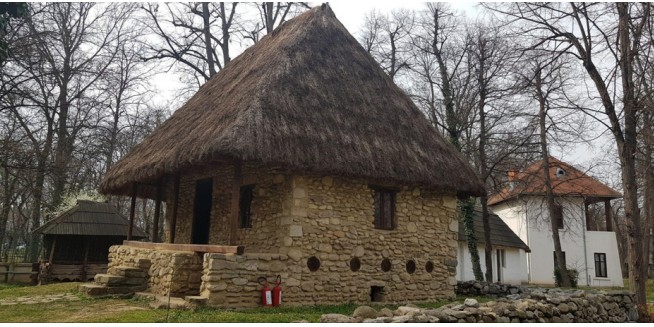

**Figure 3.** Stone walls.

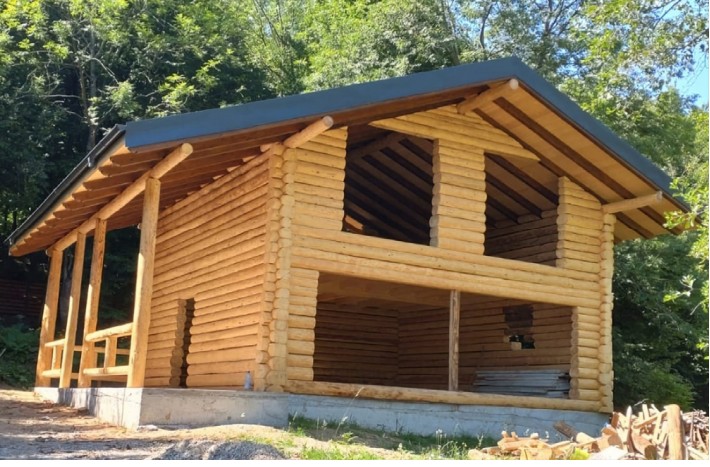

**Figure 4.** New log house in Valea Uzului, Bacău County.

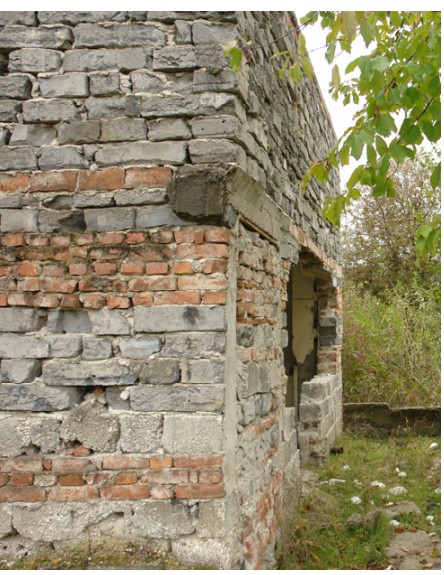

**Figure 5.** Alternated layers of stone and bricks in Voineşti, Dâmboviţa County (@ arch. Cornelia Zaharia).

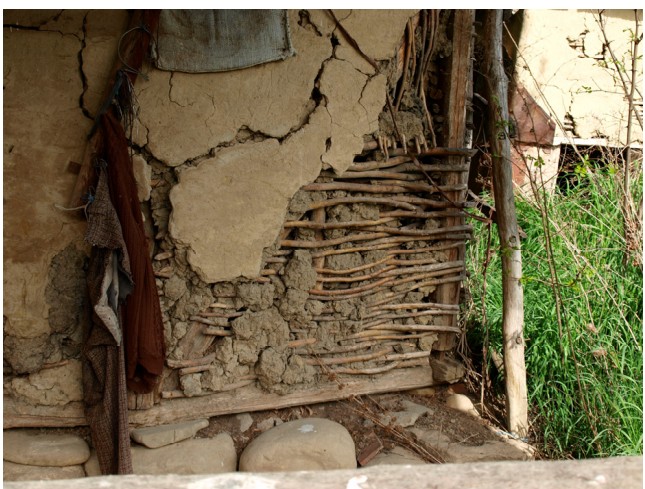

**Figure 6.** Wattle and daub house in Lunca Frumoasă, Pârscov, Buzău County (@ arch. Cornelia Zaharia).

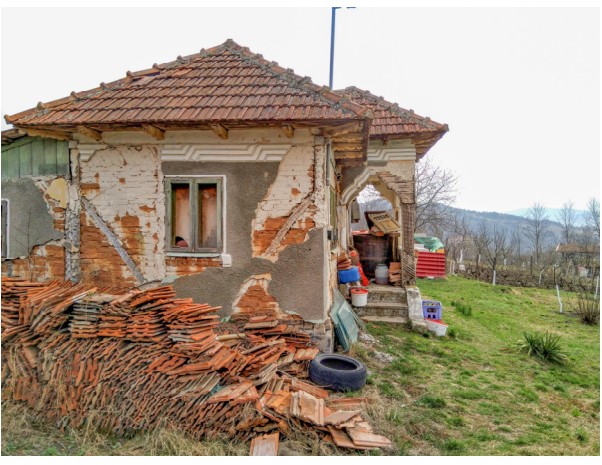

**Figure 7.** *Paianta* house with burned brick masonry infill in Lunca Frumoasă, Pârscov, Buzău County (@ arch. Cornelia Zaharia).

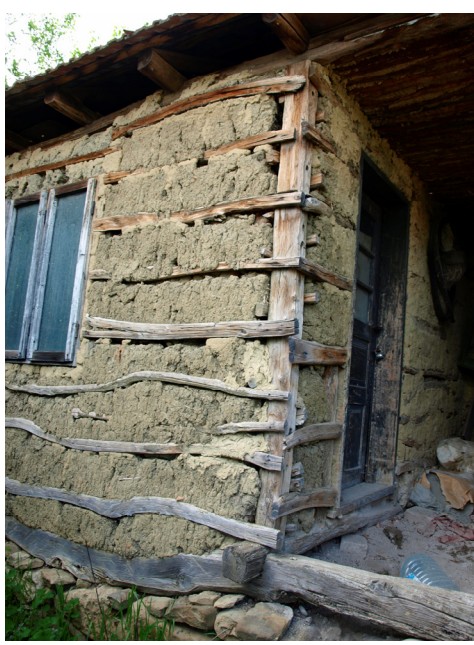

**Figure 8.** *Paianta* house with horizontal strips in Lunca Frumoasă, Pârscov, Buzău County (@ arch. Cornelia Zaharia).

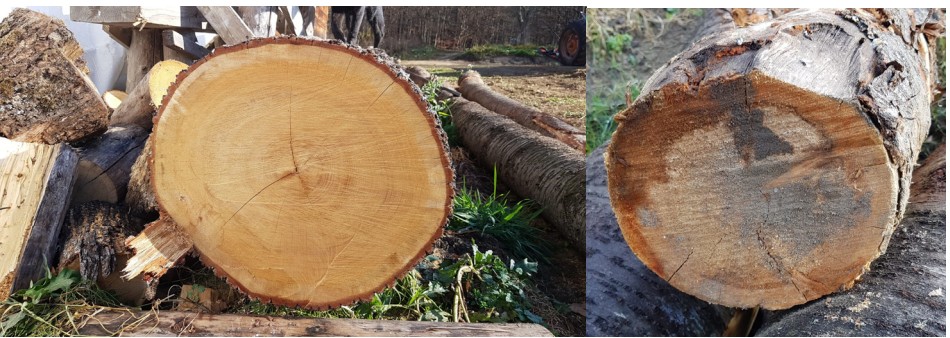

**Figure 9.** Wood for construction, good quality (**left**) and bad quality—for fire (**right**).

The connections between the posts, the top plate, the mudsill and the braces may be of mortise and tenon type, cross-halved, or just nailed.

The clay used for the infills should be yellow and as sticky as possible [20]. To obtain a more stable infill, casein (from milk) was sometimes mixed with lime and clay. Horse manure was also used to impermeabilize the surface. The mortar for the infill was made with clay, chaff, minced straws, mixed altogether by squeezing them with the feet for about six hours, preparing in one batch only the necessary quantity to be used during a day of work [20]. The mortar can be applied either as large, layered chumps, or inserted into frameworks. After placing the first layer all around the house walls, this would be left to dry for a week, then the second layer would be placed, and so on.

The bricks are made of clay which is usually collected in autumn, left to be weathered until spring, and then processed into burned bricks. The most burned clay would be than selected.

The wall envelope [18] may be made either of timber wainscot (''*lambriu*''), or of three layers of plaster made of dung and earth (2:1). The finishing is also based on lime, dung, earth, straw, sand and, more recently, cement, in different recipes. Sometimes brick or tile debris can be mixed in the finishing as well. The finishing is applied directly on the walls or on a support layer. The support may be made with the ax, chopping directly on the timber, or of timber strips (Figure 10), twigs laid in zigzag, wire mesh and nails.

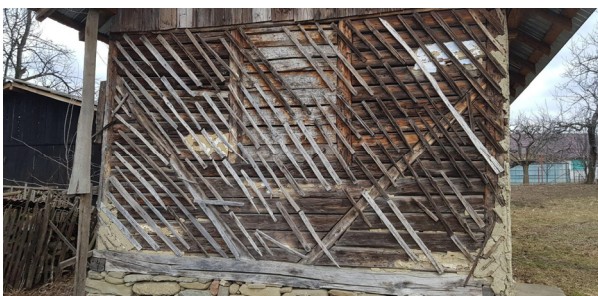

**Figure 10.** Finishing support (hooping) made of timber strips (or twigs) laid diagonally.

### 3.4. The Roof

The roof framing is made of timber elements laid in various ways, depending on the dimensions of the house, the skills of the worker/owner and the availability of materials. Thus, the rafter ends may be supported either on the top plate of the wall, i.e., on the perimeter of the building, or they can extend as an overhang, covering the porch as well. The interior frame supporting the rafters may be made of a single ridge beam, placed on props, and of a bottom beam, placed under the prop. For larger houses, typically with more than three rooms, three to ten beams can be disposed. The transversal beams ("*cordițe*") can be also found, spaced 60–80 cm apart, projecting on the exterior of the roof, as overhangs, by about 60 cm (Figure 11). The connections are made with iron/steel nails or clamps.

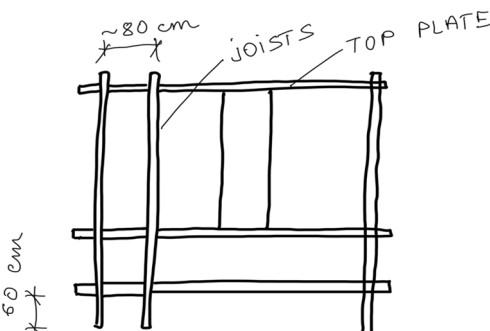

**Figure 11.** Roof framework seen from above, Păcureți, Prahova County (@Maria Elena Enăchescu) [18].

The shape may be of shed ("*într-o apă*"), gable ("*în două ape*") or hip ("*în patru ape*") type (Figure 12).

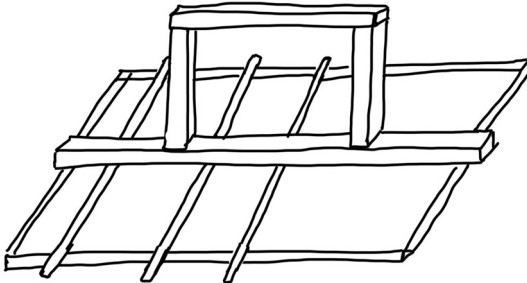

**Figure 12.** Hip roof ("*în patru ape*") framework [18].

The **roof cover** differs, depending again on the area, materials availability, local culture and labor skills. In the southeastern part of Romania (Dobrogea), where reed is easily available, the roof cover can be made of reed, or of earth or pantile ("*olană*") (Figure 13). The cover may also be made of wheat or rye ("*secară*") straws. The connection between the

straws and the roof structure may be made with some strips to keep them fixed during the first years or tied with twigs or wires. The roof cover can also be made with corn stalks or long planks. The shingle ("*șindrilă*"), sieve ("*șiță*"), and clapboard ("*draniță*") are also popular, and differ according to their dimensions; the sieve is small, the shingle is average size and the clapboard is large [20]. These latter three can be made of fir or beech, sometimes of oak. Tiles ("*țigle*") were also used, as well as galvanized metal sheets. Later, asbestos-cement sheets ("*azbociment*") were used.

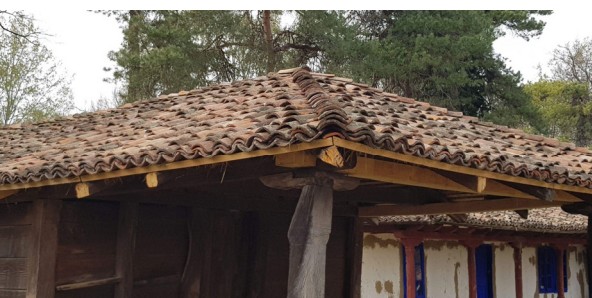

**Figure 13.** Pantile roof cover (National Village Museum).

## 4. Types of *Paiantă* Houses

The difficulty of identifying the constructive details of the *paiantă* typology comes also from the different regional terms that are used to denote them in the historical provinces of Romania. Thus, depending on the area or on the sources, the *paiantă* is defined as follows.

In Muntenia and Dobrogea [18]: 1. Construction technique for walls, with earth rammed into a twigs or planks formwork. 2. Construction material, wattle and daub or battens infilled with earth/mud. 3. Thick planks, diagonal braces used also to erect the adobe walls, by supporting the posts in order to prevent tilting/displacement during construction; they may be made of wooden beams, carved timber or planks. 4. Planks laid between the rafters, for support. 5. Reed or poplar wattle placed between the posts, to build the fences. 6. Little planks nailed to the rafters, to which the roof cover is connected.

In Oltenia, Banat, Crișana and Maramureș [21,23]: 1. Construction technique used both for houses and household annexes. Twigs, strips or battens are nailed to posts inserted into the ground. 2. Horizontal timber elements nailed on both sides of the posts, spaced 0.25–0.35 m apart. The spaces between them are filled with clay chumps mixed with straws or chaff.

In Moldavia [22]: 1. Construction element used for walls, consisting of timber strips nailed horizontally on the posts, on which, at certain distances, diagonal braces are connected. 2. Posts connected to the horizontal strips forming a fence. 3. Big nails, 20–30 cm long, sometimes made of wood.

Also, as defined in [19]: 1. Constructive system used in the southeastern Europe as a version of the "*Fachwerk*", having a timber framing made of moderate cross-section components, laid vertically, horizontally and diagonally, infilled with short planks/logs, wattle and daub, bricks, adobe and sometimes stones. The system was used in southern Romania both in villages and in towns, including Bucharest. The *paiantă* represents the diagonal timber braces.

Based on the literature review and on field observations, it was noticed that the timber braces, which are considered in this research as the specific elements to include a house into the "*paiantă*" typology, can be sometimes found also in other typologies. For example, in the wattle and daub typology there are many houses in which the main timber frame is provided with this type of braces ("*paiante*"). For this reason, in the present study it was considered that those are also part of the "*paiantă*" typology. This research focuses only on braced timber frames with infills, given that they have proven their resilience almost all over the world [38] and that they were not studied enough from a scientific perspective, in order to understand in detail their good seismic behavior.

Based on several field investigations, the following types of structural systems that may be included in the *paiantă* typology were identified:

- with timber skeleton and brick masonry infill structure (Type 1—"*paiantă*"—classic) (Figure 14);
- with timber skeleton and wattle and daub infill (Type 2) (Figures 15 and 16);
- with timber skeleton and horizontal strips, infilled with earth and straw (Type 3—called "*în grădele*" only in Buzău County, otherwise it is just called "*paiantă*") (Figure 17);
- with timber skeleton, infilled with horizontal timber planks (Type 4, Figure 18);
- with timber skeleton and AAC (autoclaved aerated concrete) masonry infill (Type 5, Figure 19). This type is actually a deviation from the traditional house and it is not encouraged due to the incompatibility of the materials.

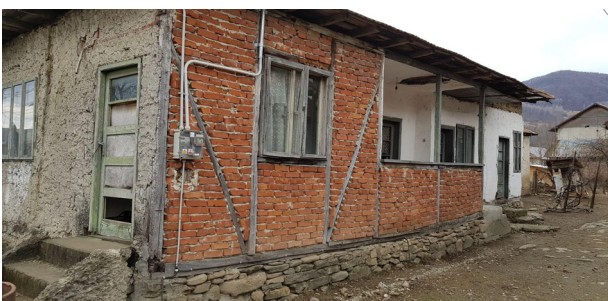

**Figure 14.** Type 1: House with timber skeleton and brick masonry infill structure in Vărbilău, Prahova (@Andreea Dutu).

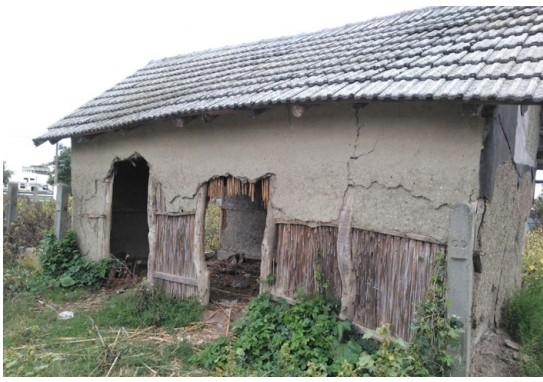

**Figure 15.** Type 2: House with timber skeleton and wattle and daub infill (vertical layout) in Sulina, Tulcea (@Marina Gingirof).

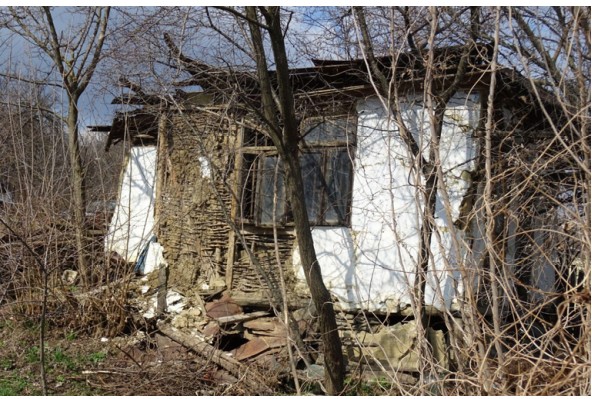

**Figure 16.** Type 2: House with timber skeleton and wattle and daub infill (horizontal layout) in Băbeni, Vrancea (@Daniel Dima).

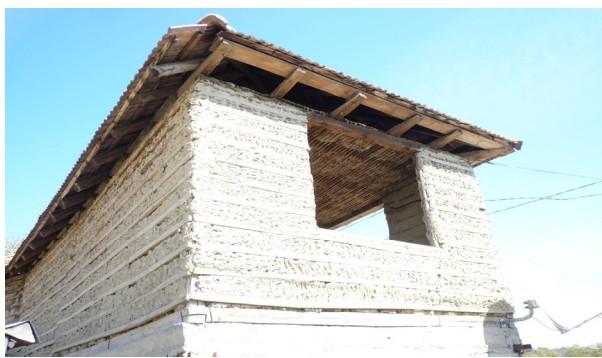

**Figure 17.** Type 3: House with timber skeleton and horizontal strips, infilled with earth and straw (@M Gingirof).

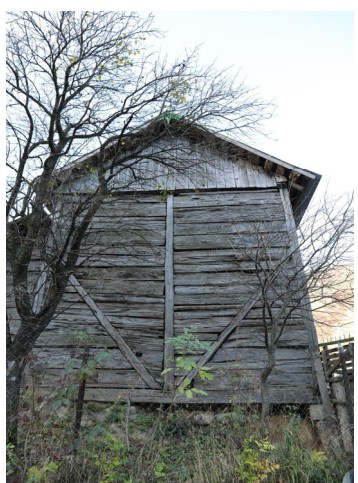

**Figure 18.** Type 4: House with timber skeleton infilled with horizontal timber planks in (@M Gingirof).

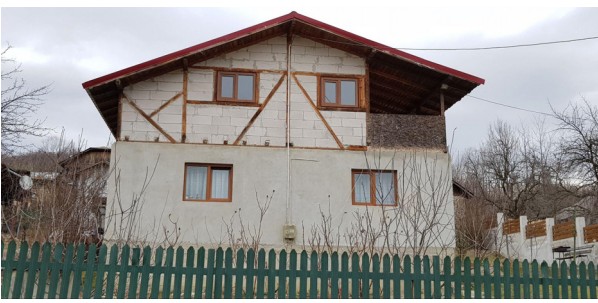

**Figure 19.** Type 5: House with timber skeleton and AAC (autoclaved aerated concrete) masonry infill in Vărbilău, Prahova (@A Dutu).

The figures show houses in a rather poor state just to present the structural system, but many of them, which are representative for the local culture, are well preserved and are covered with plaster and finishing. An example of a maintained traditional house, which is also inhabited, is shown in Figure 20, but the structural type of *paianta* is not visible.

Some of the investigated houses were abandoned, while others were well-maintained, with owners not complaining about special issues with them. Depending on the area, as noticed also during the field investigations conducted by the Order of Architects of Romania (OAR) RURAL Working Group in order to develop local architecture guidelines for traditional houses [39], certain types of *paianta* prevail, since this depends, as mentioned, on the local culture, available materials and workmanship skills. As an example, an entire

village (Deduleşti, Vrancea County) was considered in the investigation performed by the Technical University of Civil Engineering (UTCB). Here, most of the houses were investigated and a survey was conducted to assess the number of houses of each type that can be found in the village. Some of the houses with undamaged finishing the structural type could not be investigated, as owners would not allow plaster removal and/or some of the younger owners were not aware of all the constructive details of house. In this case, the type was assessed based on the similarity with other contemporary houses from the same village or ethnographic area, for which the structural type was known.

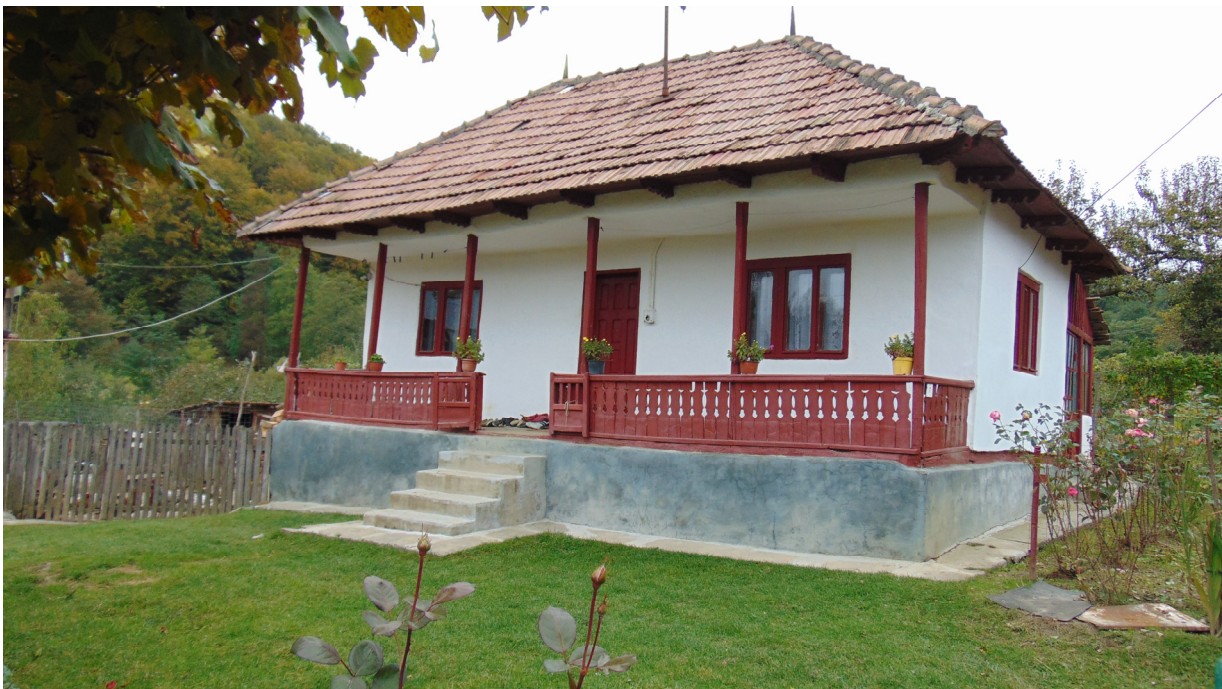

**Figure 20.** Inhabited and well maintained traditional house in Voineşti village, Dâmboviţa County (@ architect Cornelia Zaharia).

Thus, out of the 25 houses investigated in the same village, 72% were of type 1 (timber frames with brick masonry infills), 12% were of type 2 (timber skeleton and wattle and daub), 8% were of type 3 (timber skeleton and horizontal strips, infilled with earth and straw). To be noted that 12% of the houses were not identifiable, and one of the houses (Figure 21) contained parts of all the three structural types.

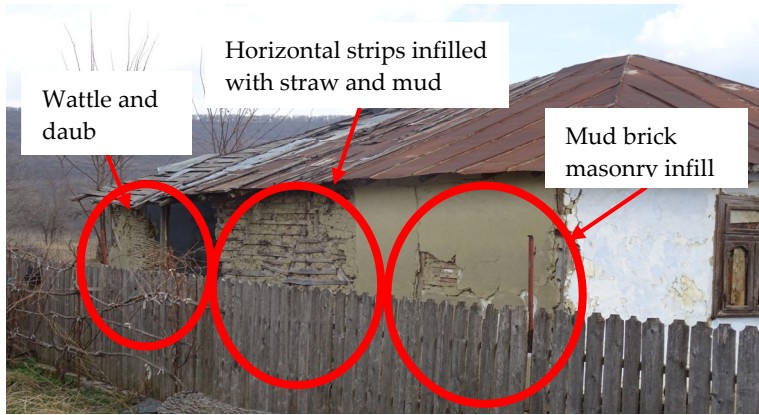

**Figure 21.** House with timber skeleton and three types of infills in Deduleşti, Vrancea (@Daniel Dima).

### 5. Most Common Degradations Found in *Paiantă* Houses

There are several types of issues, typical for traditional houses, *paianta* ones included, most of them not being produced by hazards (earthquakes or storms), but by daily use. Most of the damage and degradations that occur in almost all traditional timber houses are related to water infiltrations and humidity. These can cause several issues, all of them being included under the term of biological decay. This may be of many types, with mold as the most common. Wood-eating insects (the woodworms) are also affecting the timber structure and can lead to a spongy appearance of the timber, which actually becomes weak, crumbly and finally disintegrates. Other degradations can be caused by lichens and algae (Figure 22), which, by releasing oxalic acid, may induce significant degradations in timber elements, and also in stone socles.

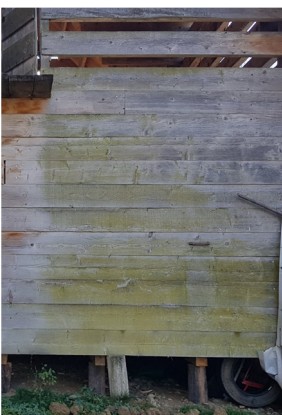

**Figure 22.** North side of a house affected by algae.

The most frequent water infiltration issues occur where the roof cover is degraded, so that the water can reach the roof framework, which degrades usually at the joints. The infiltrations can also appear at the mudsills, when these are not properly protected from the capillary water coming from the soil, or when the eaves are not large enough to protect the facades from rainwater—thus, basically, where the humidity is present near the foundations.

The next most common sources of degradation are usually the wrong conformation of the house, which can result in cracks in the walls (usually vertical) and the differential settlements which can also lead to cracks, as well as other structural deformations.

The degradations are presented further in six categories: accidental (Table 1), environmental (Table 2), mechanical (Table 3), improper conformation (Table 4), poor maintenance (Table 5), inappropriate interventions (Table 6). The degradations found in all the three case studies are included in the tables, and further presented in more detail. The symbols used in the tables to describe degradation are as follows: ○ = present; × = not present; N/A = not applicable.

**Table 1.** Accidentally produced degradations.

| Degradation Factors | Degradation | H 1 | H 2 | H 3 |
|---|---|---|---|---|
| Landslides/soil creep causing differential settlements | Vertical and inclined cracks | × | × | × |
| | Out-of-plane deformations | × | × | × |
| Earthquakes | Out-of-plane deformation | × | × | × |
| | X-shaped cracks | × | × | × |
| | Joints detachment | × | × | × |

**Table 1.** *Cont.*

| Degradation Factors | Degradation | H 1 | H 2 | H 3 |
|---|---|---|---|---|
| Floods | Enhancing biological decay | ○ | × | × |
| | Degradation of the socle masonry | × | × | × |
| Storms | Detachment of the roof cover | ○ | ○ | ○ |
| | Detachment of the rainwater drainage elements | × | × | × |
| Fire | Total or partial destruction of the timber elements | × | × | × |

**Table 2.** Environmentally-induced degradations.

| Degradation Factors | Degradation | H 1 | H 2 | H 3 |
|---|---|---|---|---|
| Heavy precipitation/stagnating water near the walls/water washing the facade of the house | Humidity from the capillary action causing: | × | × | × |
| | degradation of the socle masonry | | | |
| | mold, fungus, algae, bacteria proliferation, | | | |
| | which finally results in: | | | |
| | biological decay of mudsills and wooden facade cladding | | | |
| | plaster peeling | | | |
| | exposure of wall structure to environmental factors | | | |
| Freeze-thaw cycles, thermal shocks due to rapidly changing temperature | Cracks | ○ | ○ | ○ |
| | Spalling | × | ○ | ○ |
| | Deformations of the timber structure due to the sudden loss of moisture content, the increase of humidity relative to the decrease in temperature | ○ | ○ | ○ |
| Deposits of green algae of lichens on the surface of exterior finishes | Degradation of timber elements | × | × | ○ |
| Rodents and bird droppings | Reduction of timber elements cross-section | × | × | × |
| | Harmful bacteria proliferation | × | × | × |

**Table 3.** Mechanically-induced degradations.

| Degradation Factors | Degradation | H 1 | H 2 | H 3 |
|---|---|---|---|---|
| Poor foundation soil | Differential settlements, out-of-plane deformations | × | × | ○ |
| Undrained soil | Increased capillary action and humidity | × | × | ○ |
| Collapsible soil | Differential settlements | ○ | ○ | × |
| | | × | × | × |

**Table 4.** Degradations caused by improper conformation.

| Degradation Factors | Degradation | H 1 | H 2 | H 3 |
|---|---|---|---|---|
| Lack of foundations/insufficient foundation depth | Detachment of timber elements | ○ | N/A | ○ |
| | Cracking | × | × | × |
| | Differential settlements | × | × | × |
| Supplementary loading caused by changes in functionality or by the application of heavy finishings | Cracking | × | × | × |
| | Detachment of timber elements | × | × | × |
| | Loss of strength and stability of load-bearing walls | × | × | × |

**Table 4.** *Cont.*

| Degradation Factors | Degradation | H 1 | H 2 | H 3 |
| --- | --- | --- | --- | --- |
| Load pattern errors, inappropriate/insufficient load-bearing elements | Cracks | × | × | × |
| | Timber elements bending | × | × | × |
| Poorly designed/executed joints | Joints detachment | × | × | × |
| Timber defects | Crooked timber elements | × | ○ | ○ |
| | Holes in timber elements | × | × | × |
| | Cracks in timber elements | ○ | ○ | ○ |
| Use of already infested timber for construction | Timber disintegration | × | × | × |
| Undried timber with over 30% humidity | Crooked/shrinked timber elements | × | × | × |
| | Cracked timber elements | × | × | × |
| Use of softwood timber | Fast biological decay due to low durability of the timber species | ○ | × | × |
| Socle made of highly hygroscopic materials | Deterioration of the socle masonry | ○ | × | × |
| | Loss of foundation strength | × | × | × |
| | Biological decay of mudsills | × | × | × |
| | Superstructure dislodgement/sliding | × | × | × |
| No perimeter pavement around the house | Moss, algae or lichen films appear at the base of the walls | ○ | × | ○ |
| Insufficient eave overhangs (widths) | Socle and walls exposed to various types of degradation due to excessive humidity exposure | × | × | × |
| North-facing facades exposed to high humidity without the possibility of drying | Biological decay on north-facing facades | × | × | ○ |

**Table 5.** Degradations caused by poor maintenance.

| Degradation Factors | Degradation | H 1 | H 2 | H 3 |
| --- | --- | --- | --- | --- |
| Failing to timely intervene for eradicating a biological attack of the facade | Biological decay | ○ | ○ | ○ |
| Allowing abundant vegetation and trees to shade facades, thus maintaining increased facade moisture, roots penetrating the houses' foundation | | ○ | ○ | ○ |
| Uninhabited house, unheated, producing inside an increase of the air humidity | | ○ | ○ | ○ |
| | Cracks in the clay mortar joints | ○ | × | × |
| Lack or degradation of the rainwater drainage system—gutters and downspouts | Permanent humidity in the area of the socle and the facades (rain washing the facades), favoring the degradation of the plaster and then of the infill and timber structure | ○ | ○ | ○ |
| | Destruction of the eaves | | | |
| Lack of windows and doors | Increase of the relative humidity indoors, allowing the penetration of insects, birds, people, etc., hence biological degradations as a consequence of destructions caused by humans | × | × | ○ |

**Table 6.** Degradations caused by inappropriate interventions.

| Degradation Factors | Degradation | H 1 | H 2 | H 3 |
|---|---|---|---|---|
| Embedding original socle in concrete, attempting its retrofitting | Capillary action | × | ○ | ○ |
| | Biological decay | × | ○ | ○ |
| Improper joint strengthening | Detachment of timber elements | × | × | × |
| Additional elements or bearings changing the static scheme of the structure | Bent timber elements | × | × | × |
| | Cracks | × | × | × |
| Cement plastering on the socle and walls | Biological decay | × | × | × |
| | Plaster spalling | × | ○ | ○ |
| | Spalling of the masonry along with the cement plaster | × | × | × |
| Use of asphalt board waterproofing over timber elements | Mudsill biological decay | × | × | × |
| Mudsill placed directly on the soil or on a brick masonry foundation | Biological decay | ○ | N/A | N/A |
| Lack of socle | Biological decay of timber elements | ○ | × | ○ |
| Poorly designed interventions to low-durability mudsills | Biological decay of timber elements | × | × | ○ |
| Lack of traditional construction knowledge | Structural instability | × | × | ○ |
| Lack of knowledge on modern materials compatibility with natural, traditional ones | Biological decay | × | × | ○ |
| Use of vapor proof varnishes or paints that alter the wood structure | Biological decay | × | × | × |
| Adding waterproof thermal insulation | Biological decay | × | × | × |
| Building concrete walkways or paving tangent to the socle | Enhancement of capillary action, dampness | × | × | × |
| Existence of heavy machinery traffic roads in the vicinity of the building (inducing vibrations into the house structure) | Wall cracking | × | × | × |
| | Differential settlements | × | × | × |

## 6. Selected Case Studies

Among the five identified types of *paianta*, the first three are the most common. All three of them have been previously investigated experimentally [40] for the behavior under combined vertical and lateral loads. Only one single wall from each type could be constructed and tested, and for this reason the variation of the results could not be properly characterized, even though some interesting conclusions could be drawn. The results of the tests showed overall a ductile behavior, with the main damage consisting of cracks in the infills. The presence of braces doubled the capacity to lateral loads. On the other side, for type 1 (timber frames with masonry infills) a sensitivity of the infills in the out-of-plane direction was noticed, explainable by the fact that, even from the beginning, the mud mortar cracks due to drying shrinkage and has no connection with the timber frame. The actual behavior in earthquakes confirmed the results of the experiments and showed that, usually, even if the infills may fall, the timber frame will be able to further support the loads. It was noticed that the timber connections, although poorly executed sometimes, can have significant deformations, however not large enough to lead to the collapse of the house [38].

To illustrate the actual condition of *paianta* houses located in various regions of Romania, a number of three case studies is presented in the following. Even though there are several similar houses currently inhabited and well maintained in the investigated areas, the case studies were chosen from not inhabited and not maintained houses. The degrada-

tions from various factors were more accentuated in these buildings and the structure was partly exposed due to fallen plaster, thus easier to inspect.

The map in Figure 23 shows the locations of the case study houses. All of them are situated in rural or peri-urban areas of the extra-Carpathian part of Romania. The first two are typical for the plain and hilly zones of Muntenia, in the southern part of the country, while the third one is located near the port town of Sulina, in the Danube Delta.

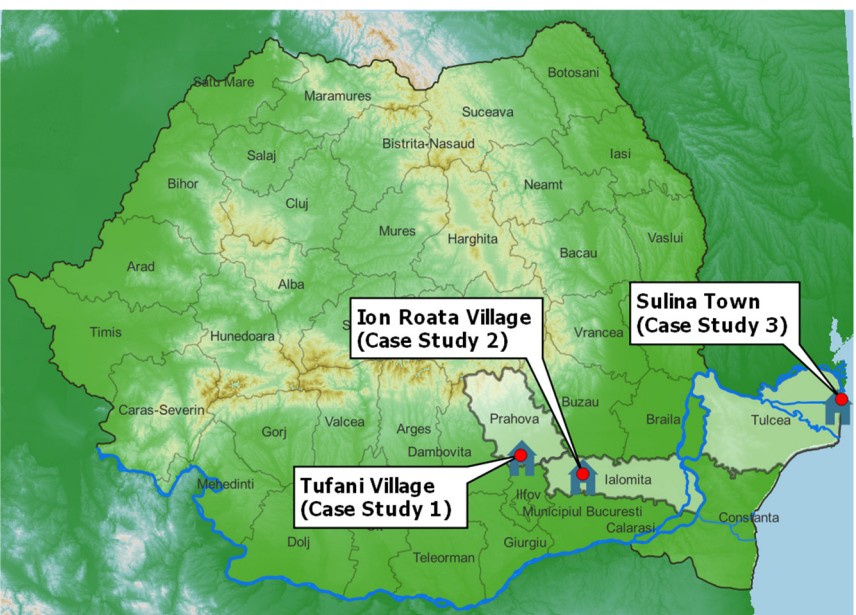

**Figure 23.** Locations of houses considered for the case studies, with local topography shown on the map.

The seismicity of the above areas is significant, as they are quite close to the Vrancea seismogenic source, located at the Carpathian arc bend. Table 7 shows the design values of the peak ground acceleration, $a_g$, and of the characteristic (corner) period, $T_C$, according to the Romanian seismic code, P100-1/2013 [35].

**Table 7.** Design values of peak ground acceleration, $a_g$, and characteristic (corner) period, $T_C$, according to the Romanian seismic code, P100-1/2013, for the locations of case study houses.

| Case Study | Peak Ground Acceleration, $a_g$ [g] | Characteristic (Corner) Period, $T_C$ [s] |
|---|---|---|
| 1. Tufani Village | 0.35 | 1.6 |
| 2. Ion Roată Village | 0.35 | 1.6 |
| 3. Sulina Town | 0.20 | 0.7 |

As it can be noticed from Table 7, all case study houses and, in particular, the first two, are located in areas characterized by rather high $a_g$ values. As regards the $T_C$ value of 1.6 s for the first two sites, these were introduced in the seismic code to account for the narrow-band frequency content, concentrated at short frequencies/long periods, observed during the past strong earthquakes in Bucharest and in large areas of the Romanian Plain. Given the short natural periods of the analyzed structural typologies, this could be also considered as one of the explanations of the absence of severe seismic damage and collapse in such buildings, as already shown in Section 2 of the paper. For the third case study, even though a design value $T_C = 0.7$ s is specified in the seismic code for the entire south-eastern

area of the country, local site effects related to deltaic alluvium could have led in Sulina to the shift of spectral peaks towards longer periods, thus to less aggressive effects of strong earthquakes for this short-period house typology.

A detailed description of the seismicity of Romania, including the $a_g$ zoning map according to the P100-1/2013 code, can be found in [41].

Regarding the climate for the locations of the studied houses, this is characteristic to that of the Romanian Plain (Lower Danube Plain) for the first two case studies and to the Danube Delta for the third. The climate of the Romanian Plain is temperate-continental, with the average annual temperature ranging from 11–11.5 °C in the south to 10.5° in the north [42]. The mean rainfall is about 500 mm/year [42]. The climate of Sulina, in the Danube Delta, is continental, with strong influences from the vicinity of the Black Sea. The climate is characterized by an average annual temperature of 11 °C, with the highest average temperature of 22 °C in July and the lowest average temperature of −1 °C in January [43]. The mean rainfall is about 350 mm/year. According to the Köppen climate classification, the first two case study houses are located in Dfb areas (wet temperate continental), while the third is located in a Dfa (wet warm continental) area [44].

### 6.1. Case Study 1 (Prahova County, Tufani Village, No. 14)

The building (Figure 24) is only one story-high (ground floor) with the components described as follows.

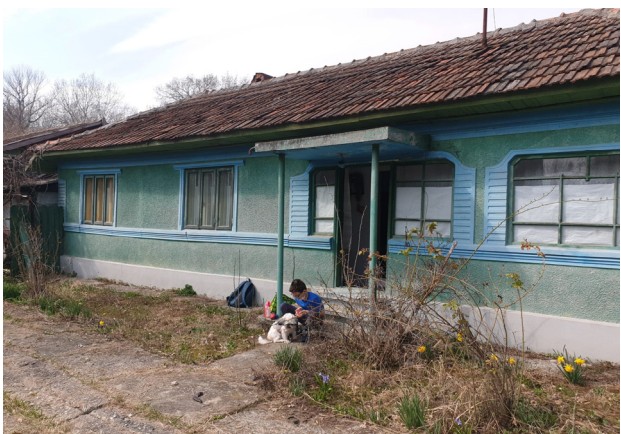

**Figure 24.** House 1, Tufani Village, Prahova County.

The house has a hip **roof**, made up of rounded and peeled timber elements, joined by means of clamps and nails. The rafters of the roof truss are made of 12 cm-diameter round timber, spaced at about 90–100 cm (Figure 25). On the rafters there are battens with a section of 3.8 × 5.8 cm, spaced at about 25–30 cm apart, which support the tile covering of the roof. The tile covering consists of ceramic tiles, about 40 × 22 cm in size, produced at Cărpiniş Brick Factory.

The **superstructure** of the building is made up of load-bearing walls arranged both transversely and longitudinally, which are made in two constructive variants. At the top, the walls of the structural system are connected by means of a floor made of wooden joists oriented in the short direction of the building (spaced about 60 cm apart) and plank footing. A network of tangent elements, made of roughly processed wood pieces (with 7 × 5 cm cross-sections) (Figure 26) is fixed to the soffit, perpendicularly to the direction of the joists, forming the ceiling structure. The ceiling supports the thermal insulation layer, inserted between the slab beams, made of a rubble filling mixed with clay.

On some areas of the floor surface, the **ceiling** is made of wooden paneling (about 20 cm wide each) nailed to a network of roughly processed tangent elements (Figure 27). On the soffit of the floor above the entrance hall (between axes 4–5/A–B) (Figure 28), the ceiling is made of plaster with a wire ("rabitz") mesh (Figure 29).

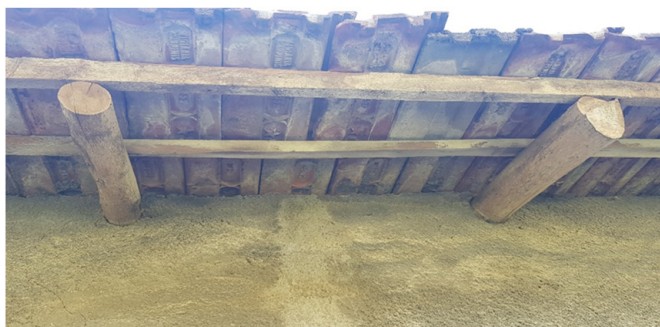

**Figure 25.** Roof cover supported on the batten network that transmits loads to the rafters.

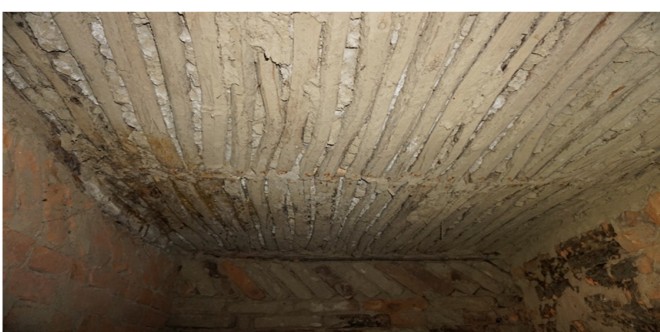

**Figure 26.** The floor soffit over the ground floor. The network of roughly worked pieces of wood fixed to the intrados of the beams is visible.

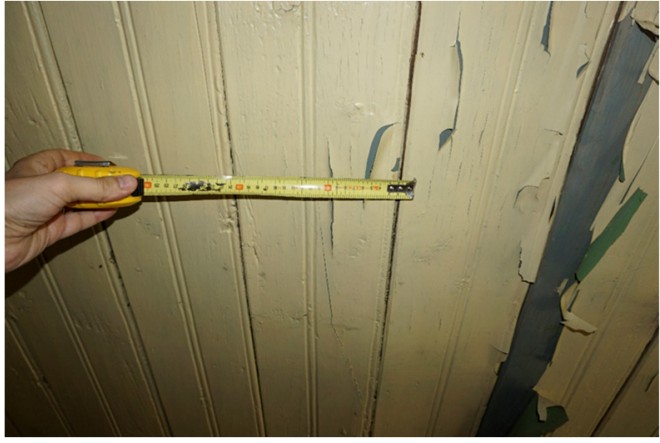

**Figure 27.** Wainscot fixed to the soffit of the floor in certain areas.

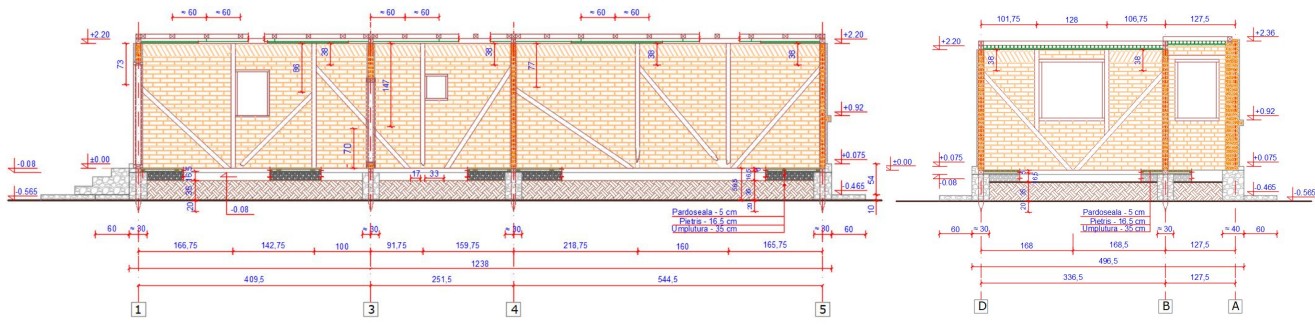

**Figure 28.** Lateral views of the house; (the layers in the figure are: flooring—5 cm, gravel—16 cm, earth fill—35 cm).

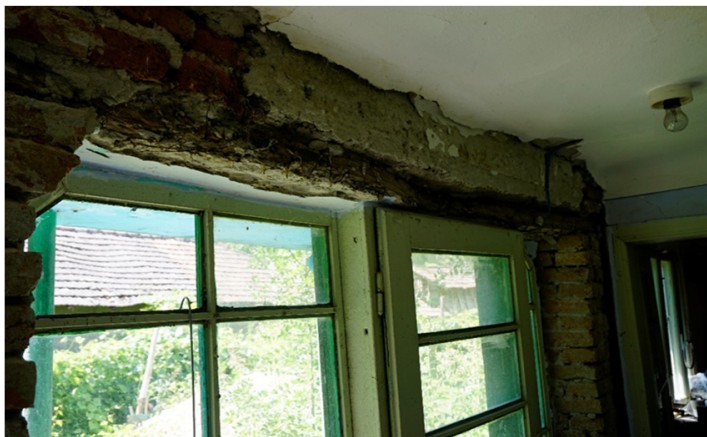

**Figure 29.** Ceiling in the hallway area.

The envelope walls in axis A and between axes 4 and 5, respectively in axis 5 and between axes A and B, are made of plain burned clay solid brickwork (240 × 115 × 63 mm) with a thickness of one brick (Figure 30). The rest of the walls of the house (placed on both directions) have a mixed structure, consisting of timber frames with braces (made of 8 × 8 cm pieces of squared timber) and infilled with solid clay brick masonry (240 × 115 × 63 mm) with a thickness of $\frac{1}{2}$ brick (Figure 31).

The timber frames in the structure of these walls are made of posts placed at the ends and in some intermediate locations, top plates supported on the posts (and also on the infill masonry), mudsills arranged between the marginal posts (supported on the so-called foundation) and inclined bracing elements connecting the lower boards to some posts. The posts found in the intersection areas are embedded about 20 cm into the ground at the base of the walls. The height of the wall system, including the thickness of the upper floor slab, is about 2.40 m.

Except the window openings located in the one-brick thick walls bordering the entrance hallway on the outside, where timber lintels were observed at the top, in the rest of the window openings (in the mixed structure walls) no lintels are present, and the masonry is applying loads directly onto the wood window frames (Figure 32).

In the case of walls with timber frame, the infill masonry is laid along the height of the panels, with the exception of the last portion under the top plate, where the ceramic elements are arranged in an inclined position (Figure 33).

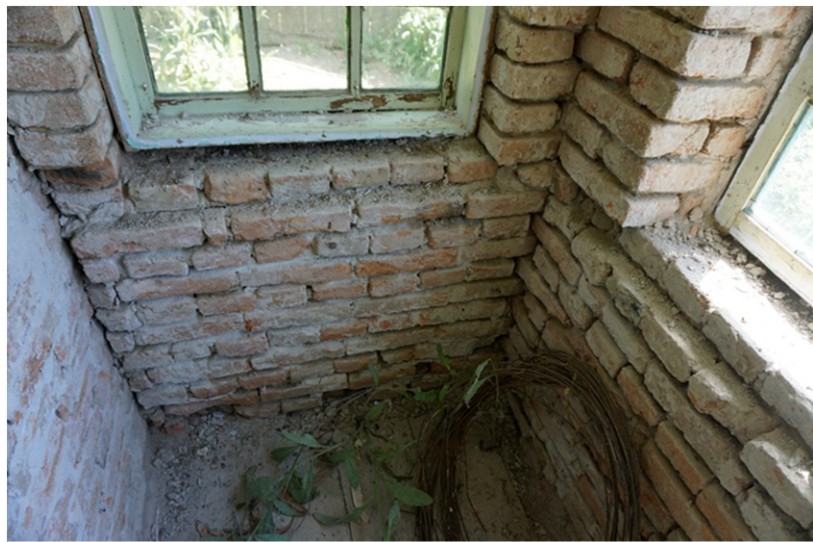

**Figure 30.** Envelope walls of one-brick thick masonry.

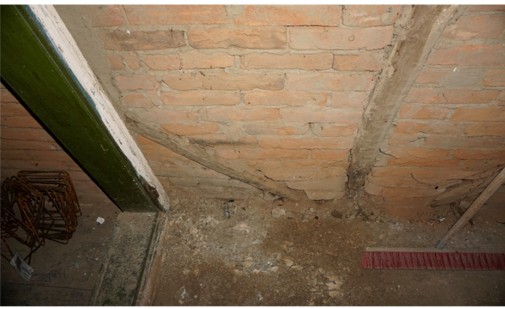

**Figure 31.** Walls with mixed composition, having a frame structure made of pieces of squared timber, braced, and $\frac{1}{2}$ brick masonry infill.

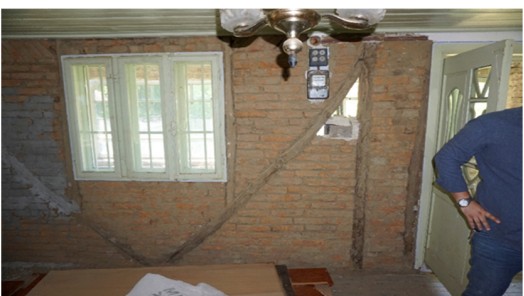

**Figure 32.** Window opening in a wall with mixed structure. The absence of the lintel above the gap can be noticed.

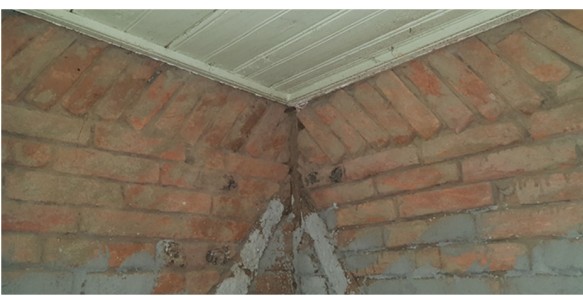

**Figure 33.** Arrangement of bricks at the top of the masonry panels.

The walls of the building are plastered with mortar, with a thickness of about 1.5 cm on the inside and 2.5 cm on the outside. To ensure the adherence between the mortar and the wooden elements of the wall structure, nails were driven into them, and wire connections were made (Figure 34).

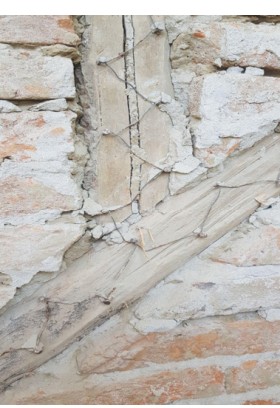

**Figure 34.** Nails and wires for an improved connection between the plaster and the wall.

The base of the walls (serving as a foundation) is made of plain low-class concrete with a height of about 50 cm and a width of about 30–35 cm for walls with a thickness of $\frac{1}{2}$ brick and about 40 cm for walls with a thickness of one brick. This concrete is cast directly on the surface of the uncovered ground, supporting the perimeter walls, as the construction does not have a real foundation. In the space bordered by the concrete beams/elements, a compacted earth fill of about 35 cm, on top of which a 16 cm-thick layer of capillarity gravel is laid (Figure 35).

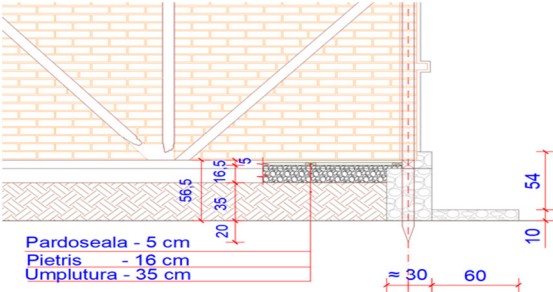

**Figure 35.** Detail showing the fixing of the wooden posts (at the intersection of the walls) at the level of the simple concrete plinth (base) and, respectively, the layers disposed in the space bordered by the plinths (the layers in the figure are: flooring (*pardoseala*)—5 cm, gravel (*pietris*)—16 cm, earth fill (*umplutura*)—35 cm).

The perimeter sidewalk, made of plain concrete, is about 60 cm-wide.

The configuration of the structure showed a ductile behavior in previous seismic events. No typical seismic damage was observed, but only degradations due to water infiltrations and lack of maintenance. Even with the timber skeleton partially degraded and the brick masonry infill also partially damaged, the house is still able to withstand seismic motions. The key feature of this type of *paianta* is the timber frame flexibility, and the infills capacity to prevent excessive deformation of the frame. The two component materials are generally not tightly interconnected; thus, through the cracks in the mortar joints of the infills and by the deformations of the timber connections (which usually have gaps from the poor execution), the earthquake energy is dissipated and the house resists seismic actions. Another very important structural feature is the presence of the braces, which have a significant contribution to the overall capacity (almost doubling it, as it resulted from experimental tests [40]), although they are connected below the upper corner joint of the timber frame.

### 6.2. Case Study 2 (Ialomiţa County, Ion Roată Village)—Type 3

The building is one story-high. Its structural components are further described.

The house has a hip **roof**, made up of rounded and peeled timber elements, joined by clamps and nails. The **roof covering** is made of ceramic tiles with a size of about 40 × 22 cm, resting on battens (Figure 36). The roof of the building extends above the porch area (about 1.20 m wide) located on the main facade (entrance area) and on the lateral sides of the house. The roof is supported on wooden joists with a section of 7.5 × 10 cm, which transfer loads to wooden posts made from the same assortment. There are no gutters on the perimeter of the roof served by downspouts, to direct the rainwater flow towards the courtyard, which is why the water running off the roof splashes on the vertical surface of the perimeter plinth, exposing it to dampness.

The **superstructure** of the building consists of load-bearing walls placed both transversely and longitudinally and built in the "*paianta*" construction technique.

The structure of the walls is made of timber frames (posts with the section of 10 × 10 cm or 7.5 × 10 cm (Figure 37), also used in other investigated locations, and beams with the same cross-section, placed on top of the posts), braced with wooden pieces of the same type. It is worth noting that the wooden elements from which the frames and bracings

were made were obtained either by sawing or by hewing (by carpenters). Elements made of round wood (hazel branches) with a 2.5–3 cm diameter are fixed horizontally (by nailing) on each side of the frame, about 10–15 cm apart (Figure 38).

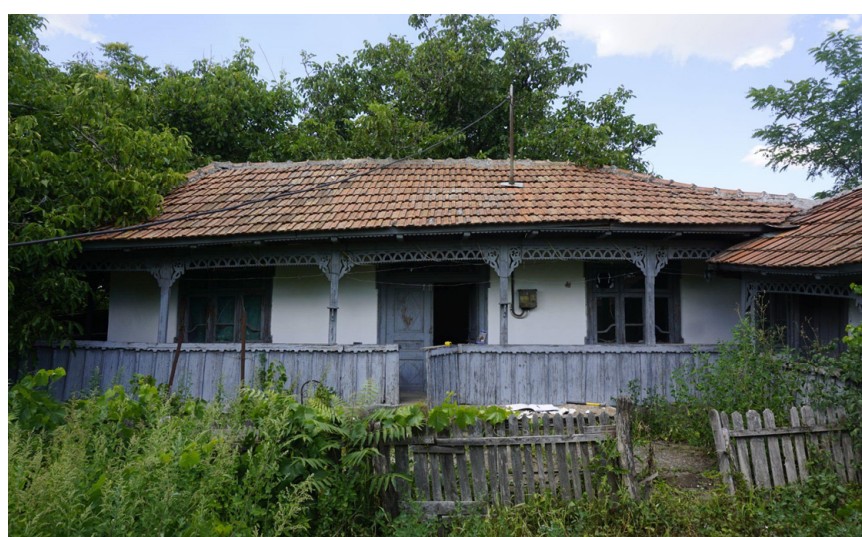

**Figure 36.** House 2, Ion Roată Village, Ialomița County.

A mixture of clay, straw and water (and occasionally horse or cow dung) was pressed into the thickness of the frames, i.e., into the space between the horizontal round timber elements, and this formed the **infill** (with thermal insulating properties) of the wall structure. On both sides of the **walls** there is a layer of plaster, with thicknesses of 2 to 2.5 cm, made of mortar (lime or lime-cement). The thickness of the walls built in this way is about 20 cm, and their height between the floor and the ceiling (i.e., the lower surface of the planks resting on the joists—the floor soffit) is about 2.40 m. At the top, the walls of the structural system are connected by means of a framing made of timber joists with a section of 7.5 × 10 cm, oriented in the short direction of the building and spaced at about 80–85 cm, supported on the walls, and of the plank decks resting on the joists (placed in the perpendicular direction) (Figure 39). The posts are inserted into the socle at the base of the walls (over the full height of the plinth), i.e., driven into the ground about 20 cm-deep. A vertical section through the building structure is shown in Figure 40, while its plan layout is shown in Figure 41.

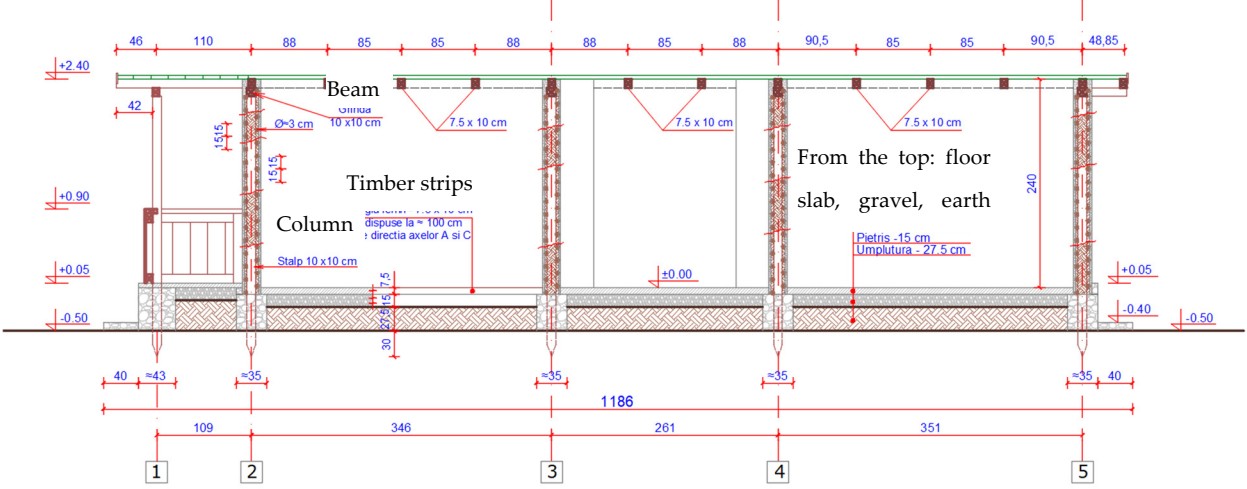

**Figure 37.** Vertical section through the building structure.

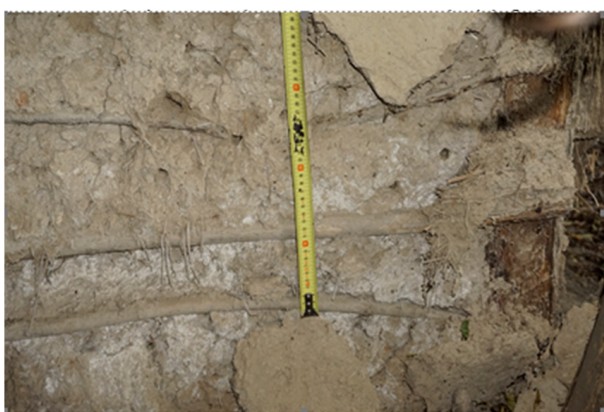

**Figure 38.** Arrangement of the horizontal wooden elements in the wall.

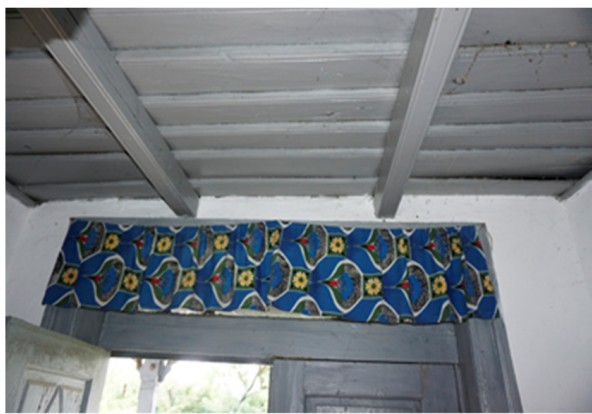

**Figure 39.** Layout of the wooden joists and of the floor deck above the ground floor.

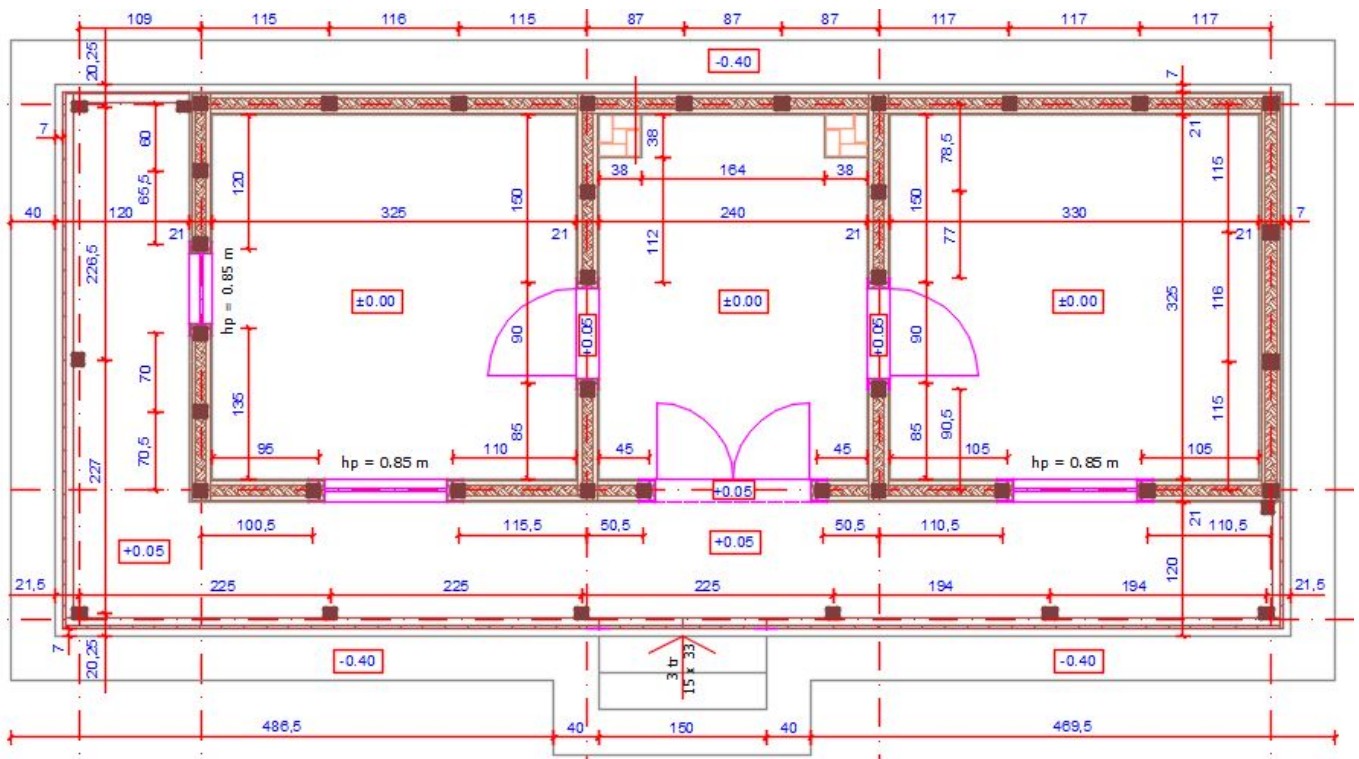

**Figure 40.** Plan layout of the house in Ion Roată Village (Ialomiţa County).

The **socle** (Figure 41) at base of the walls is made of low-class plain concrete, with a height of about 65 cm above the ground surface. The socle is also extended in the porch area, with a height of about 55 cm and the width of 30–35 cm. The plinth is built on the surface of the uncovered ground, as the construction does not have a foundation. In the space bordered by the wall socles, i.e., the one bordering the porch, a compacted earth fill is made, over which a layer of capillary break gravel is laid.

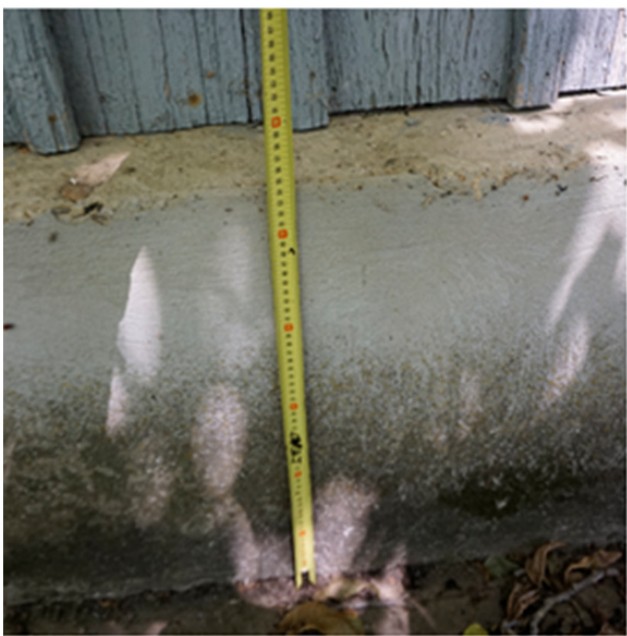

**Figure 41.** The socle at the edge of the porch.

The floor of the building is made of a layer of concrete poured between the wood elements (beams of the same type used for the floor beams/posts on the porch perimeter, and which support the roof), on which the floor planking (now removed) was initially fixed (Figure 42).

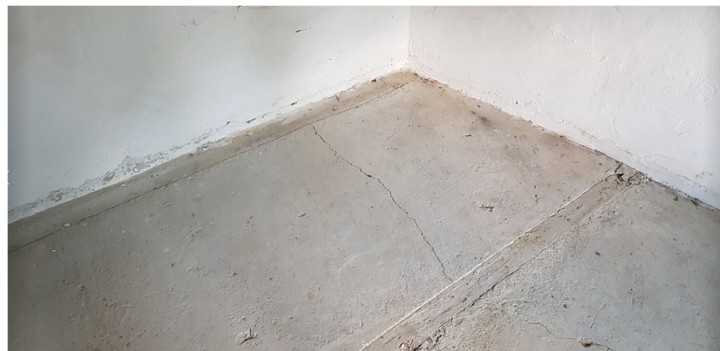

**Figure 42.** Concrete slab, poured between the wooden beams that supported the old floorboards.

The perimeter pavement is about 40 cm wide, made of plain concrete.

The seismic behavior of such a *paianta* house strongly depends on the presence of the diagonal braces. For this particular case study, they were missing, so most likely the capacity of the walls to lateral forces was reduced. However, as it resulted from the experimental analysis [40], the entire structural system is very ductile, allowing large displacements at the top with no visible or irreversible damages. The horizontal strips have a low bracing effect for the timber posts. Being very flexible and only attached with nails, their role is merely to keep in place the earth and straw mix. The seismic energy is

dissipated through the cracks in the earthen mix, and also through the timber joints of the frame.

### 6.3. Case Study 3 (Sulina)—Type 2

The house (Figure 43) has a **hip roof** with a simple framework made of rafters which are supported by props and connected with nails. The **roof cover** is made of steel sheets. However, specific for such houses are pantile roof covers. It is worth noting that there are no gutters continued by downspouts, to lead the rainwater away from the walls, thus the socle (bottom part of the walls) is always exposed to humidity. The vegetation is present just near the socle and, together with the absence of the perimeter pavement, this maintains a high humidity which, in time, causes the decay of the timber plank finishing (Figure 44)

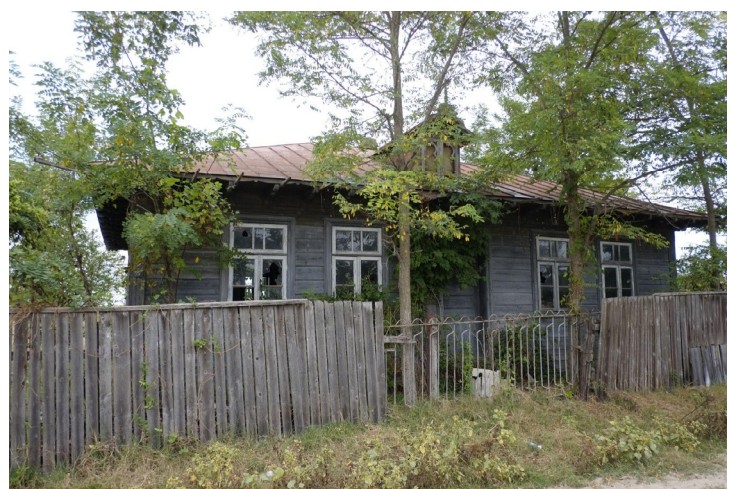

**Figure 43.** Case study 3: house in Sulina, with timber posts and a wattle and daub (laid vertically) infill.

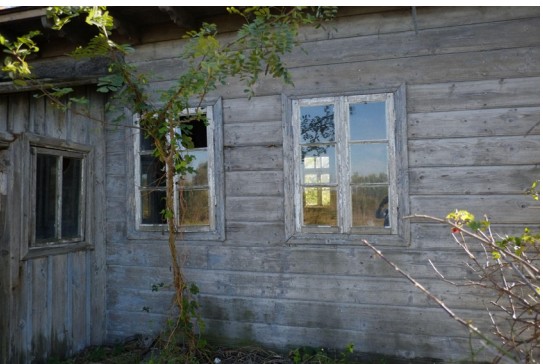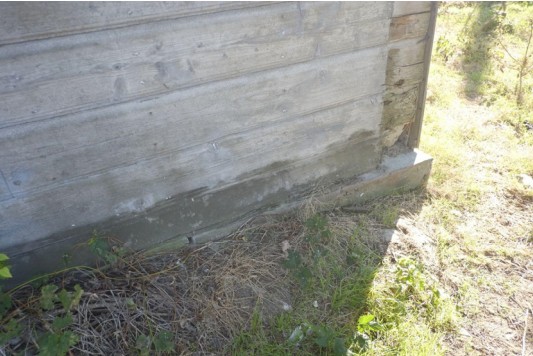

**Figure 44.** The bottom part of the plank facade is affected by dampness.

The **superstructure** consists of **walls** made of wooden posts (with about 12 × 12 cm cross-section), set into the ground and infilled with wattle and daub. They are confined at the upper part with a ring of wooden beams supporting the ceiling joists. All the walls are load-bearing and are placed in both orthogonal and transverse directions. Between the posts, along the height of the wall, several horizontal timber strips are connected with nails to the posts, on both the exterior and interior sides of the wall. Between the strips, wattle is placed vertically. This is usually connected to the horizontal strips with wires or ropes. A layer of clay mixed with chaff and chopped straws is applied on the wattle. Figure 45 presents the structural system, visible in one of the abandoned neighboring houses.

The **finishing** of the wall is made of mud mortar mixed with straw, applied in a rather thick layer (about 5 cm), on which exterior horizontal timber planks (about 20 cm wide) are

applied. This system represents the envelope of the house. The total thickness of the walls is 24 cm (Figure 46). The posts are not sawn. The distance between the posts is about 1 m, and the distance in the vertical direction between the strips is also 1 m. Thus, a "squared" geometry was created for the timber elements.

Inside the building, the **finishing** is original, made of mud and straw mortar (Figure 47). The floor decking was laid directly on the sandy soil specific to the area. In the same figure, the soil is visible, since the floor was removed in an attempt to retrofit the house.

The **foundation** of such houses is usually missing, as the **posts are set directly into the ground** at a certain spacing and a horizontal strip is applied at 10 cm above the ground, to support the twigs.

However, for this house, it seems that a concrete socle was added later on the outside part of the house, in an attempt to strengthen the structure (Figure 48). In fact, this was an inappropriate intervention, enhancing the capillary action, which further damaged the planks of the facade and the bottom of the posts.

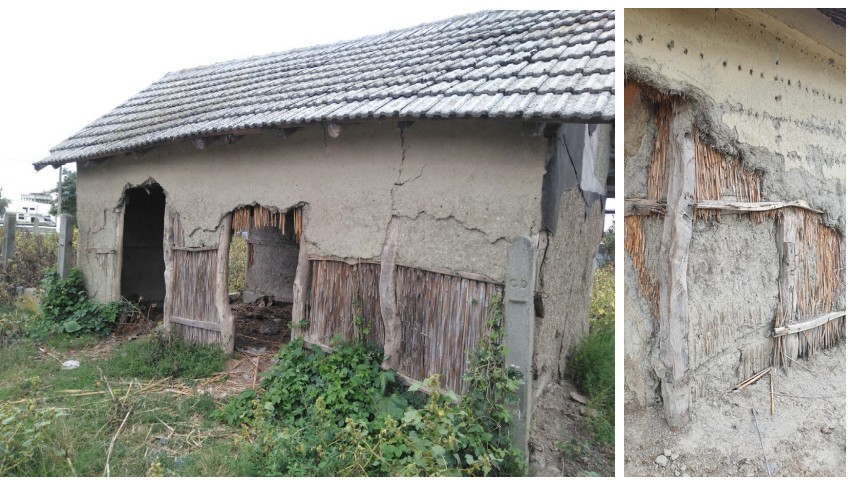

**Figure 45.** Neighboring abandoned houses, where the finishing is degraded and the inner structure is visible.

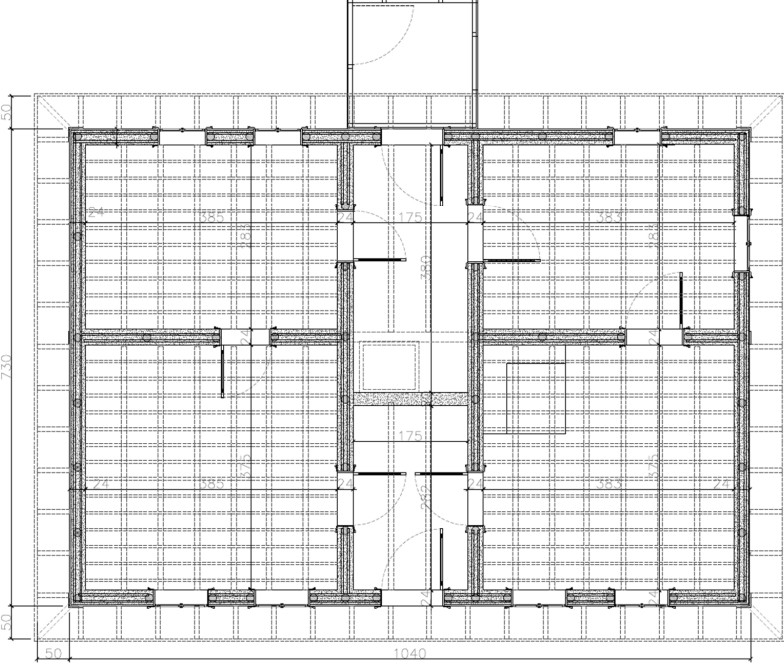

**Figure 46.** The plan layout of the house in Sulina.

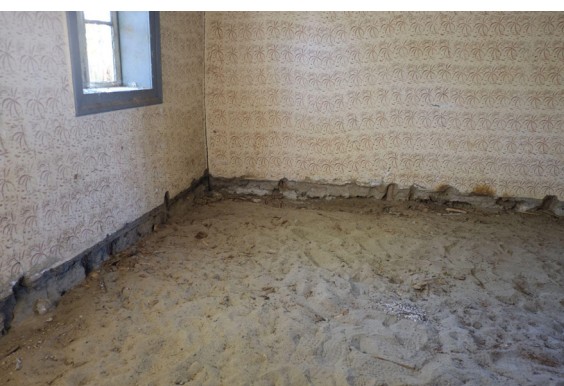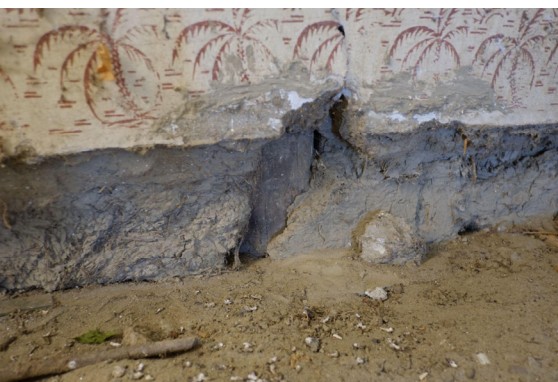

**Figure 47.** The missing floor decking reveals the earth and straw infill between the posts at the bottom part of the wall plaster.

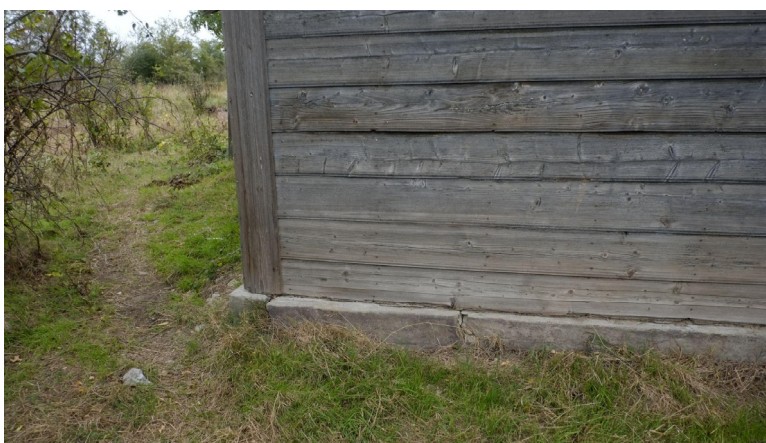

**Figure 48.** Concrete socle (retrofitting tentative).

The **slab** over the ground floor is made of timber joists spaced about 1 m apart. Between them, reed twigs are inserted and horizontal timber strips (Figure 49, left) are applied about 30–40 cm apart, to support the twigs and the clay plaster of the ceiling. This is basically the same structure as that of the walls.

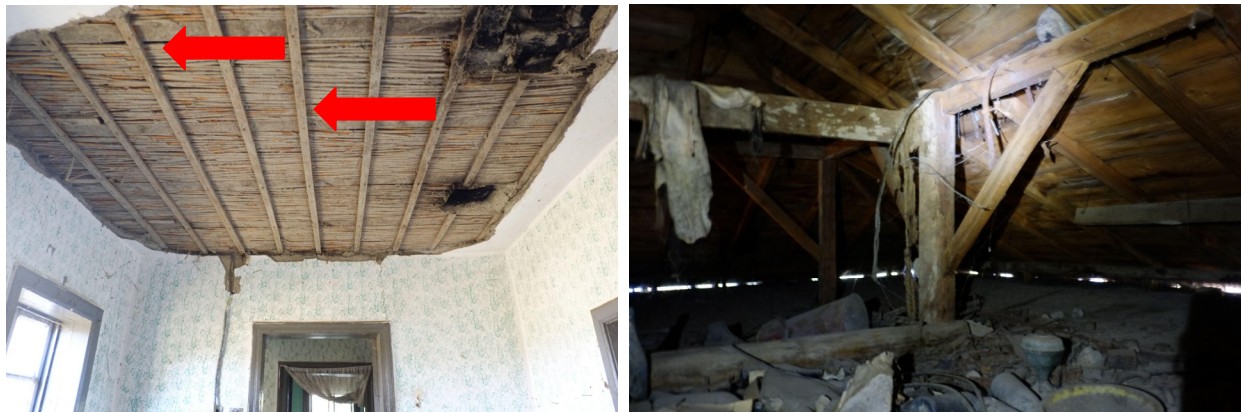

**Figure 49.** Slab over the ground floor (on the **left**) and roof framework (on the **right**).

The seismic behavior of this house type would also depend on the presence of the diagonal timber braces, which are missing here as well. Nevertheless, like in the previous case study and as resulting from the experimental analysis [40], the system, even if in this case the twigs are laid on the vertical direction, is very ductile and allows quite large

displacements at the top, with no visible or irreversible damage. The seismic energy is dissipated through the cracks in the infills, the twigs 'connections to the frame and also through the timber joints of the frame.

## 7. Observed Degradations for the Case Study Houses

### 7.1. Case Study House 1

This old building is quite degraded and not well-maintained, being uninhabited for at least seven years and affected by flooding at least once during its lifetime. Damaged, displaced or missing tiles (Figure 50) were observed on the roof covering, which allowed infiltration of water inside the building.

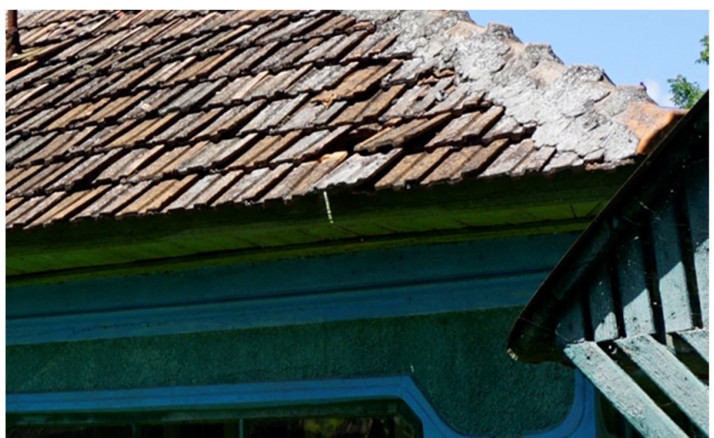

**Figure 50.** Missing roof tiles.

In the envelope walls located on the facade of the main entrance—axis A (Figure 51) and, respectively, on the lateral side facing the street—axis 5, slightly inclined 1.5–2 mm-wide cracks can be observed (Figures 51 and 52), starting from the corners of some window openings and continuing their path down to the sidewalk level. The cause of these cracks is the displacement of the ground beneath the base of the walls due to repeated freeze-thaw cycles. At the same time, the crack formation was also facilitated by the fact that the lower base on which the masonry was built had no tensioning effect, due to the local rotting of the wooden elements exposed to moisture migrating from the ground (there was no waterproofing system between the mudsill and the socle). Moreover, in the history of the house, flooding occurred and the whole house was affected by significant water infiltrations.

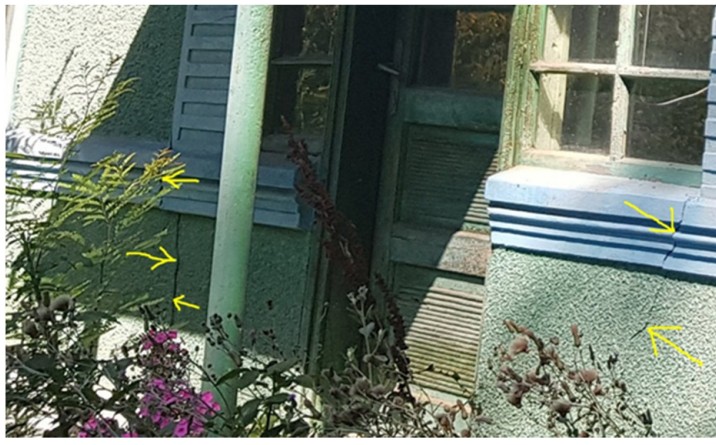

**Figure 51.** Cracks in the A-axis envelope wall, in the immediate vicinity of the entrance door. The path of the crack can be distinguished, starting from the corner of the window opening.

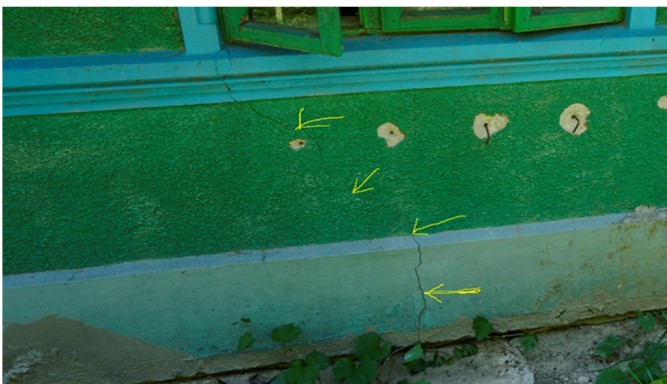

**Figure 52.** Crack in the envelope wall located in axis 5 (street-side).

The 2 . . . 3 mm-wide cracks observed in the envelope wall in axis A (the wall with the main entrance door) can also be observed from inside the building and in the vicinity of the intersection of axes A and 1 (Figure 53), which indicates a penetration of the wall. This clearly underlines the degraded state of the wall base.

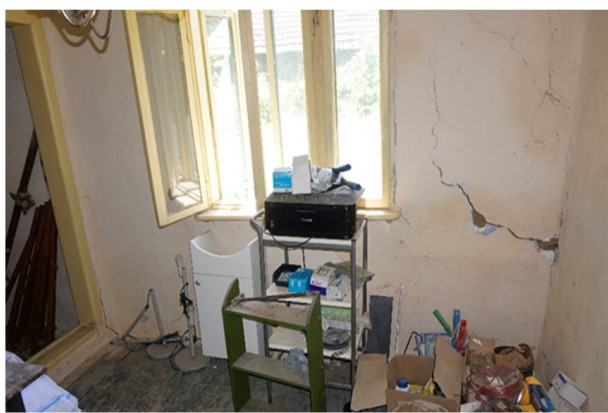

**Figure 53.** Crack in the A-axis wall, just near its intersection with the 1-axis envelope wall.

Cracks were also observed in the envelope wall in axis 1, extending over the entire height of the wall and crossing the window opening (Figure 54). Wall cracking, as a result of the settlements occurring under the socles, can also be observed inside the building, specifically in the wall in axis C (to the right of the passage door) (Figure 55), as well as in the wall in axis 2, above the door frame (Figure 56).

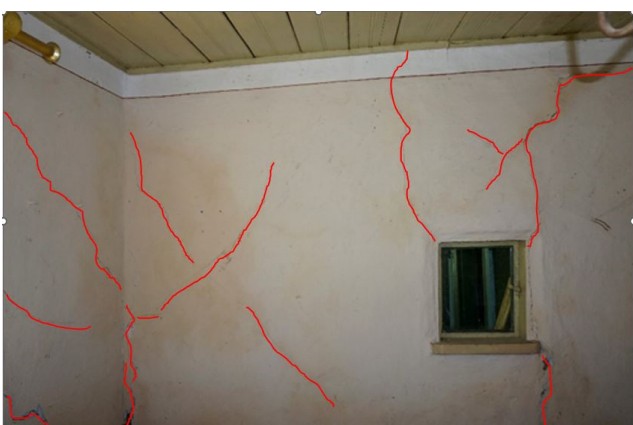

**Figure 54.** Envelope wall cracks in axis 1.

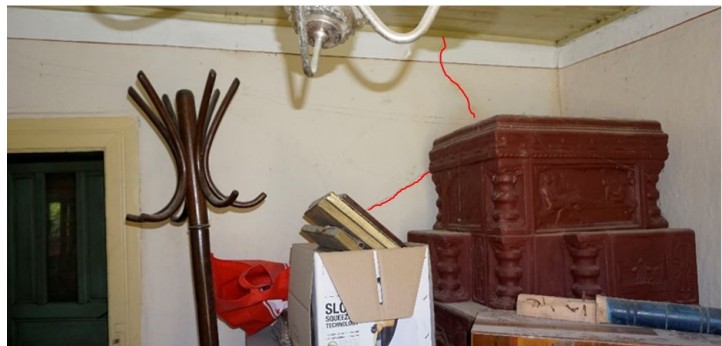

**Figure 55.** Inclined crack in the inner wall in the C-axis.

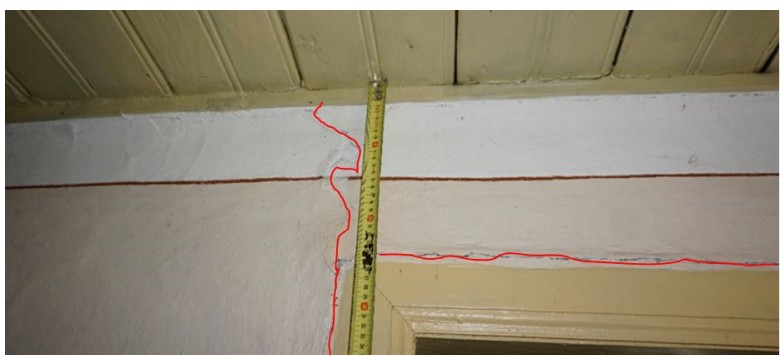

**Figure 56.** Vertical crack at the level of the inner wall in axis 2, just above the passage door.

Also, 0.5–1.0 mm-wide cracks were observed, having an inclined path, at the top of the envelope wall located in axis D (the wall behind the house), near the corner located at the intersection of axes 5 and D. It is possible that the crack is affecting only the plaster. The cause of the occurrence of these cracks is the combined effect of the possible infiltration of water into the structure of the mentioned wall (as a result of the local degradation of the coating), respectively of the subsidence produced in the ground under the plinth (the causes of which have been presented before) (Figure 57).

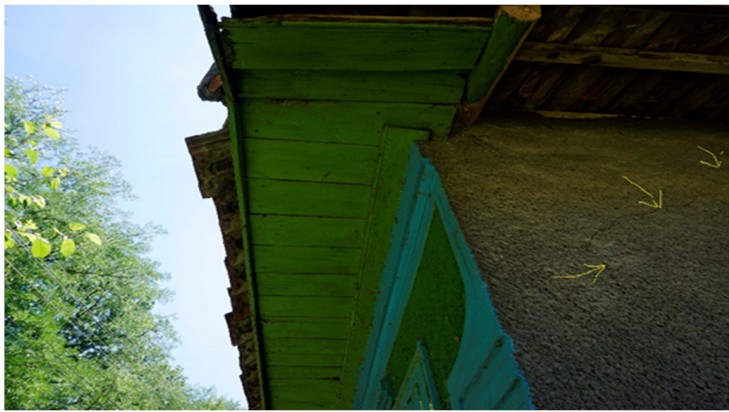

**Figure 57.** Crack at the top of the envelope wall located in the D axis (behind the house).

In some locations at the base of the envelope wall in axis D (the rear side of the house), portions where the plaster has fallen including the heavily exfoliated plinth concrete (in the immediate vicinity of the sidewalk) were observed, and the wooden elements of the wall structure (posts and mudsill) were affected by rot, their spans being interrupted. Also in these areas, the masonry of the wall is loosened locally, the frames being displaced (Figures 58 and 59).

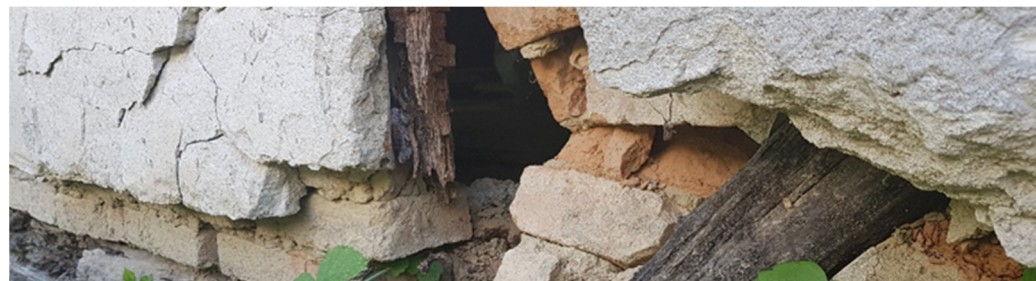

**Figure 58.** Masonry breaking in areas where segments of wooden elements embedded in the structure of the envelope wall in the D-axis have disappeared as a result of rotting.

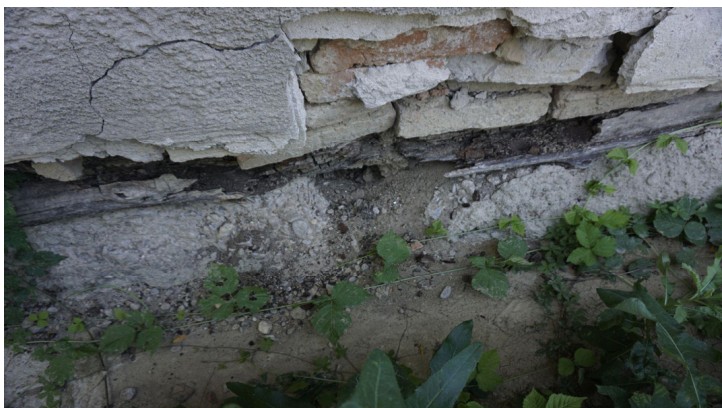

**Figure 59.** Rotting damage to the mudsill at the base of the envelope wall in axis D. The affected plaster is visible, as well as the masonry.

The damage observed on the outside of the envelope wall in axis D (the back of the house) can also be observed from the inside of the house, penetrating the entire wall thickness (Figure 60).

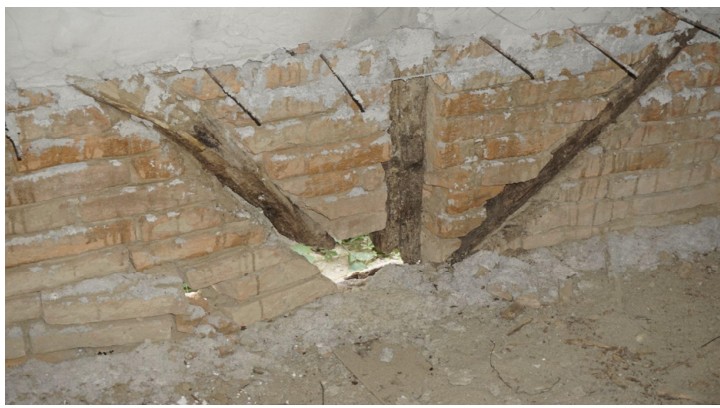

**Figure 60.** Rotting damage to the wooden elements embedded in the envelope wall structure behind the house (axis D). The affected masonry can be observed.

The lack of a foundation system under the walls of the building, combined with the lack of waterproof insulation at the base of the walls or with the ineffective measures adopted in this regard during the construction, led over time to deformations in the wall structure, thus to additional stresses, as well as to infiltrations that extended upon larger areas. Their combined effect caused the occurrence of the mentioned degradations.

Concrete exfoliation was observed in the access stairs of the building (the facade in axis A), as a result of repeated freeze-thaw cycles (in a wet state) (Figure 61).

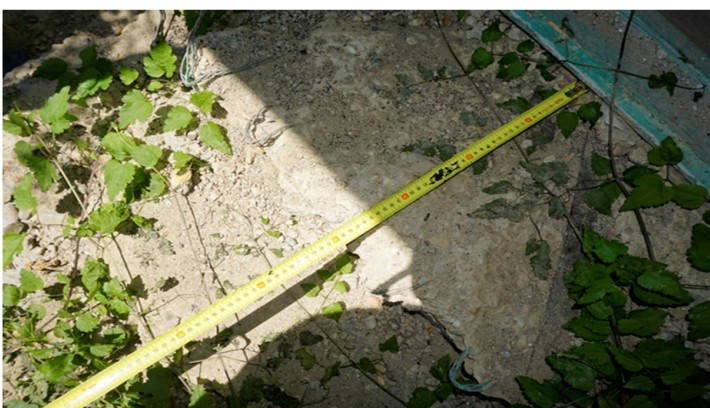

**Figure 61.** Damage in the concrete of the access staircase steps.

*7.2. Case Study House 2*

As the previous one, this house is also uninhabited, so some of the degradation is due to this fact. Part of the planks composing the **eaves** are damaged (Figure 62).

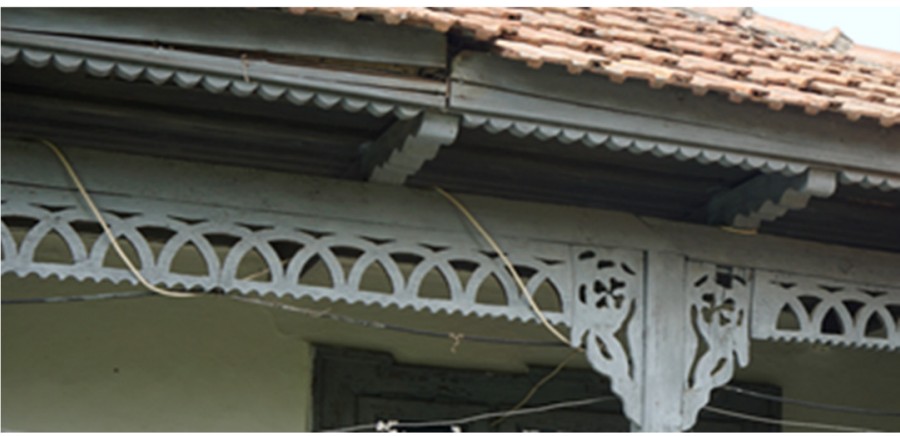

**Figure 62.** Damaged fascia at the eaves.

In some locations at the base of the walls (observation made inside the house, in a corner area) there are damp-stained areas and degraded plaster, an indication of the existence of infiltrations near the base of the walls, due to inefficient waterproofing, in conjunction with the lack of a foundation system (Figures 63 and 64).

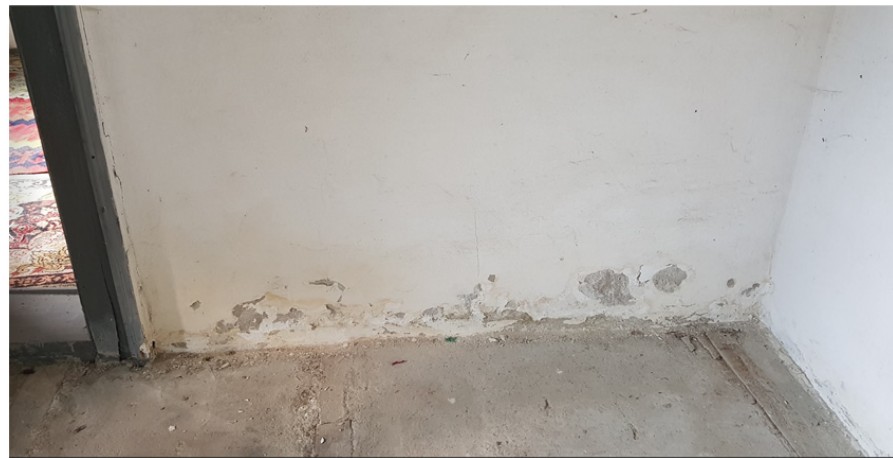

**Figure 63.** Plaster and wall finish, damaged by moisture in the wall structure.

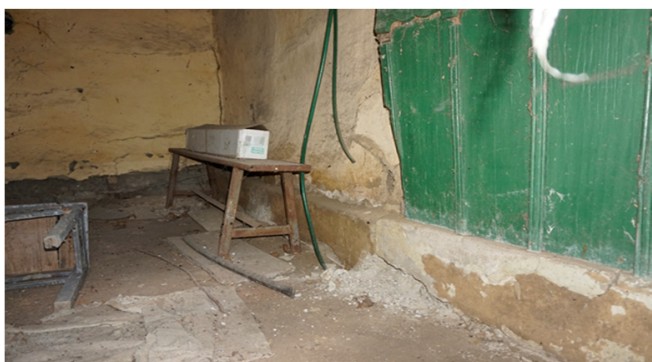

**Figure 64.** The plaster of the back wall of the house, degraded by moisture and exposure to freeze-thaw cycles in a wet state.

Also, at the rear wall of the house, on the outside, there are areas of damp decay and locations where the plaster at the base of the wall, including the plinth area, is spalling and looks like crumbling material. This appearance indicates that these areas have been exposed to freeze-thaw cycles in a wet state (Figure 64).

At the outer concrete floor of the **porch**, cracks caused by concrete shrinkage during drying (as a result of faulty execution) and areas degraded by exfoliation due to freeze-thaw cycles can be observed (Figures 65 and 66). The plaster layer applied on the vertical surface of the plinth (made of plain concrete) is locally detached due to the same reason.

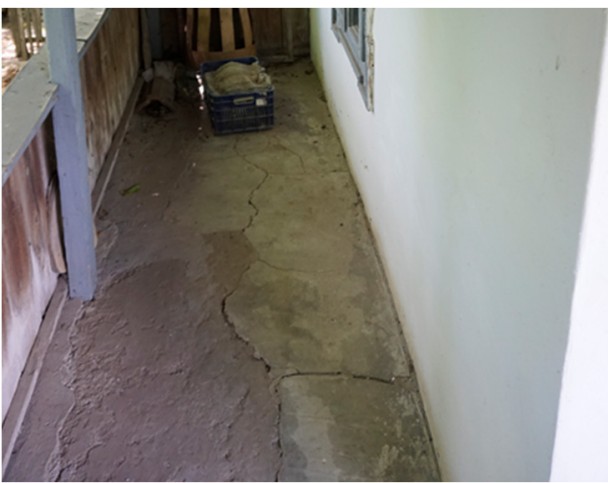

**Figure 65.** Cracks in the porch floor and areas of concrete peeling.

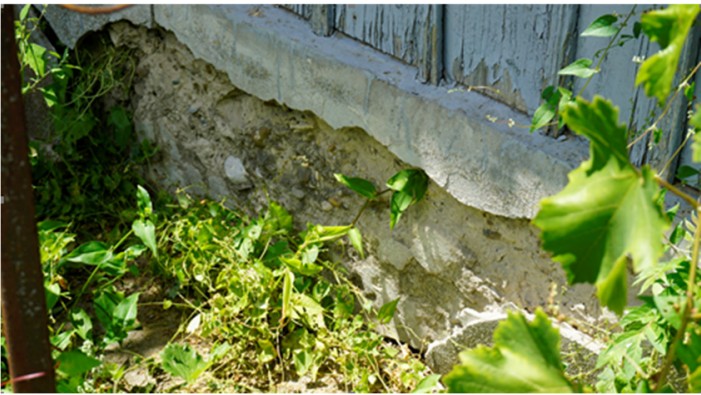

**Figure 66.** Plaster detached from the vertical surface of the perimeter plinth.

The paint applied on the surface of the wooden elements of the porch railing structure, as well as on the posts and beams supporting the roof edge in the porch area, is peeling in many places.

*7.3. Case Study House 3*

Due to humidity exposure and capillary action enhanced by the concrete socle, the post is **decayed at the bottom part**, at one corner of the building (Figure 67). Water infiltrations through the **roof** also caused the slab above the ground floor to decay.

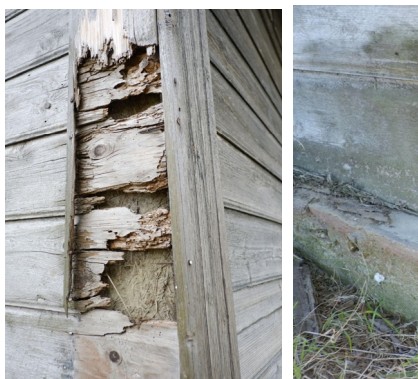 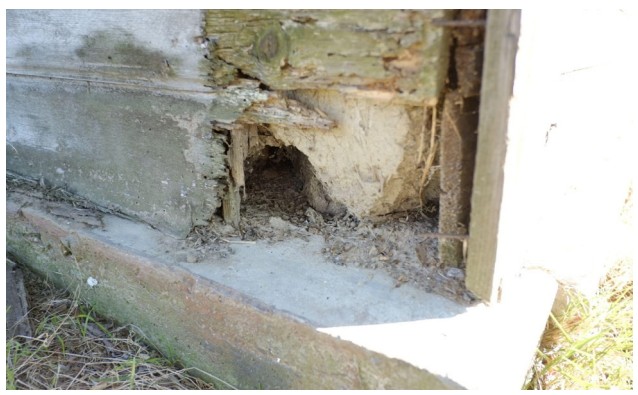

**Figure 67.** Degradation of the exterior plank facade and corner post due to biological decay.

## 8. Discussion

The three case studies presented in this paper have shown that local mold and dampness are some of the most common degradation-causing factors. These appear due to water infiltration, mainly caused by poor maintenance. All the considered houses are not inhabited; thus, they are not heated during the cold seasons, which also favors excessive indoor humidity. It should be noted, however, that poor maintenance generally leads to similar effects, no matter the building age or type.

House 1 has structural problems already due to the advanced biological decay of the mudsill, and it needs replacement of the decayed timber elements and dismantling of the existing brick masonry, which is not difficult due to the weak mud mortar. Then, the infills will be restored using the dismantled bricks. This work needs to consider additional support for the roof framework. Besides this, foundation underpinning is necessary to stop the differential settlement producing cracks in the walls.

Among the studied cases, the second house appears to be less affected by infiltrations, although more biological tests are required to determine if mold and fungus is already present. With minimum interventions to replace the degraded roof cover on the backside, filling the cracks from the backside wall with earth and straw mix, replastering it and ensuring constant heating during winter, the house can easily become inhabitable again.

The repairing process proposed for House 3 aims to restore the house to its initial structural state by removing the inappropriate additions and replacing the deteriorated parts. According to [20], the restauration works will consist of giving the structural elements their original strength, within the prevention and conservation objective. The restoration process allows structural additions, such as timber braces, when proved necessary by the structural evaluation done by an expert. Some of the most common works will be replacing the roof cover, repairing the affected post and the façade in the corresponding area, removing the concrete socle, creating a sidewalk and adding gutters continued by downspouts to drain the water away from the bottom part of the house.

Unfortunately, more and more traditional houses are abandoned in Romania nowadays, for various reasons. Often, after local owners have passed away, their inheritors, having moved a long time ago to urban areas, either disregard the houses for their obsolescence, or they completely ignore the way of retrofitting them (or do not afford to do it),

in order to provide the comfort and safety according to modern needs (indoor bathroom, thermal efficiency, water and sewage facilities, non-conventional sources of energy etc.). An example of what happens to abandoned houses is shown in Figure 68, where the same house was photographed five years apart, in 2017 and 2022. The front right corner of the house failed, dragging the roof with it, which led to a general collapse. The failure reasons may be either the biological decay of the mudsill or of the bottom of the corner post, or some differential settlements of the foundation. Whichever the case, this collapse could have been avoided by timely retrofitting measures.

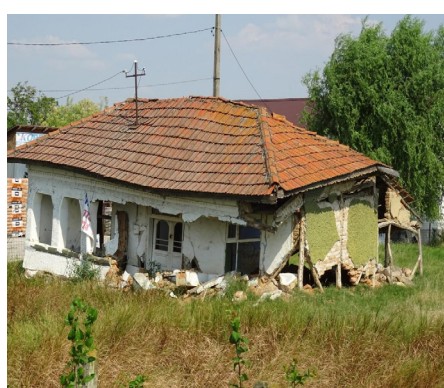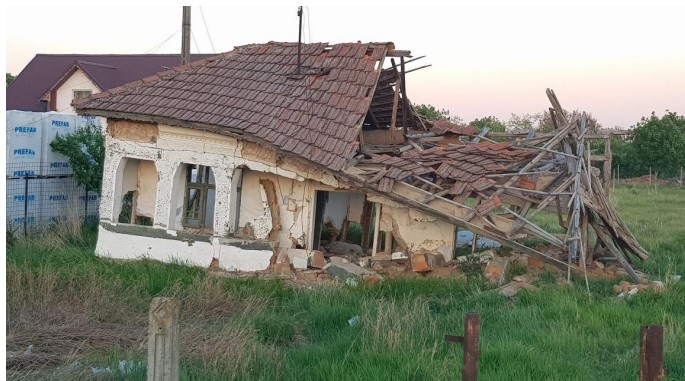

**Figure 68.** Costeşti, Buzău County, August 2017 (**left**) and August 2022 (**right**).

When compared to the popular typology of nowadays, reinforced concrete frames with masonry infills, the *paianta* houses may need more maintenance in time. Because the concrete buildings appeared more recent than the traditional ones, most of them have concrete foundations, which generally prevent differential settlements that affect many *paianta* houses. However, even concrete houses, if not inhabited and well maintained, in time can also develop dampness inside. From the seismic point of view, in certain cases, like in the 2010 Haiti earthquake, the traditional houses showed a more resilient behavior and, when the infills collapsed, the timber frame still stood up, supporting the roof. Such a behavior usually prevents human life loss, unlike the poorly constructed concrete houses that produce many casualties.

## 9. Conclusions

Traditional timber frames with infills, when properly built and maintained, can represent cheaper and sometimes healthier alternatives to concrete or masonry housing, and can even be used for reconstruction with locally available materials in disaster affected areas. Moreover, they are generally earthquake resilient, as past seismic events showed, which is an essential feature in an earthquake-prone country like Romania.

A scientific approach, combining engineering and architectural perspectives, is necessary to properly understand and re-valuate the local culture and craftsmanship, the specific construction details and materials. Furthermore, deeper analyses of mechanical and thermal properties are needed, along with experimental and numerical testing, to enable a rational and modern approach in the retrofitting of existing traditional buildings of this type and in building new houses that would fructify the old craftsmen knowledge.

This paper briefly presents the traditional houses typologies in Romania, highlighting significant details on the construction methods and illustrating them by an engineering evaluation of three houses/case studies, all representative for the same typology (*paiantă*), but having different component materials and layout, according to the specific tradition of the areas in which they are located. The observed degradations were also described, and a synthetic table was provided, showing all possible issues in traditional timber framed houses, with a special focus on those actually observed in the three case studies. These tables can be used in future field investigations, including them in the template

investigation sheets and helping specialists to check directly for the specific problems that are common for this typology.

The findings provided by this research are necessary, due to the lack of detailed information on this topic, especially from the engineering perspective. Moreover, the few available info is often misleading, not being supported by scientific methods.

The paper aims to inform and support the next steps of the ongoing research: experimental tests and numerical analysis that will also consider appropriate strengthening techniques for *paiantă* houses. The engineering scientific perspective has to take into account the materials compatibility and the seismic behavior of the house. Thus, the selected strengthening and repair methods should consider a balance between local traditional methods and the contemporary ones, based on the use of newer materials. The benefits of various solutions should be weighed, and the intervention decision should take into account all the involved criteria.

One of the most important drivers of the present study is that the Romanian villages are in danger of totally losing their local identity, due to the aggressive reconstruction or to the demolishing of traditional houses, which are frequently in bad condition. To counteract this phenomenon and to recover and re-valuate the traditional expertise and craftsmanship, an essential question has to be answered: are modern houses better than the old traditional ones in terms of health, safety, sustainability and durability? Further research within the current STRONGPA project, the continuation of the TFMRO project [28], will deal with these questions.

Based on the detailed case studies and also on previous research [28], numerical models will be adapted for type 1 houses, starting from [45], while for the other types they will be further developed. Also, future experimental studies on *paiantă* walls will determine the variation of their response under combined lateral and vertical loads, with several specimens being built and tested. After testing, some of the walls will be strengthened using various simple methods and then tested again, to assess the efficacy of the applied retrofitting techniques.

**Author Contributions:** Conceptualization, A.D.; methodology, M.G.; software, M.N.; validation, I.-G.C. and M.G.; formal analysis, A.D.; investigation, A.D., M.N. and M.G.; resources, A.D. and M.G.; data curation, I.-G.C.; writing—original draft preparation, A.D.; writing—review and editing, I.-G.C.; visualization, M.G.; supervision, M.G.; project administration, A.D.; funding acquisition, A.D. All authors have read and agreed to the published version of the manuscript.

**Funding:** This research was funded by the Romanian National Authority for Scientific Research and Innovation, CNCS – UEFISCDI, grant number PN-III-P2-2_1-PED-2021-1428.

**Institutional Review Board Statement:** Not applicable.

**Informed Consent Statement:** Not applicable.

**Data Availability Statement:** Not applicable.

**Acknowledgments:** This work was supported by a grant of the Romanian National Authority for Scientific Research and Innovation, CNCS—UEFISCDI, project number PN-III-P2-2_1-PED-2021-1428", acronym STRONGPA "Appropriate strengthening methods for the Romanian traditional "*paiantă*" house". The authors also acknowledge the kind collaboration of the owners of the investigated houses, Eng. Costin Târşoagă and Eng. Marius Stanciu.

**Conflicts of Interest:** The authors declare no conflict of interest.

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
