# Peer review of "Construction Techniques and Detailing for Romanian Paiantă Houses: An Engineering Perspective"

_sustainability, doi:10.3390/su15021344_

Round 1
Reviewer 1 Report
This study provides an overview of the traditional building construction techniques in Romania including its structural detailing and observed degradation for the given case studies. The overall manuscript is good and well presented. However, there are some comments to improve this paper as listed below:
1. It is recommended to explain further (with example or using the building from the case study) the effect of structural configuration and detailing on the performance of the building against seismic load.
2. It is recommended to provide a close-up view of the crack photograph for figures 33, 34 and 35.
3. Figure 18 and 42 are not clear. It is strongly recommended to amend those figures that only highlighting the relevant structural component or gridline.
4. It is recommended to include method used for observation of degradation. Any standards or guidelines used as references?
5. The observed degradation for each case study can be included in section 5 so it is easier to be compared.
Author Response
We thank the reviewer for the kind and insightful comments and suggestions, which substantially contributed to the improvement of our manuscript. We have done our best to address all the requirements. Please find below the description of the revisions that we have made. These have hopefully brought the manuscript to a level that would fully meet the reviewer’ expectations.
For an improved clarity and readability, we have also reordered and partially restructured the sections of the paper. Thus, we have created a clearer outline of the paper, as follows.
- Introduction
- Seismic performance of traditional Romanian houses
- Traditional houses: components and materials
- Types of paiantă houses
- Most common degradations found in paiantă houses
- Selected case studies
- Observed degradations for the case study houses
- Discussion
- Conclusions
We have also made a revision of the English language and of the general formatting. Small interventions were made at a few locations in the text to improve clarity and readability. All the changes were made by using the “Track Changes” feature of Microsoft Word, according to the Editors’ instructions.
We provide also a “clean” manuscript, without the “Track Changes” marks.
Comment 1: It is recommended to explain further (with example or using the building from the case study) the effect of structural configuration and detailing on the performance of the building against seismic load.
Answer 1: We added a paragraph for each case study (at the end) explaining the seismic performance of the buildings:
House 1: “The configuration of the structure showed a ductile behavior in previous seismic events. No typical seismic damage was observed, but only degradations due to water infiltrations and lack of maintenance. Even with the timber skeleton partially degraded and the brick masonry infill also partially damaged, the house is still able to withstand seismic motions. The key feature of this type of paianta is the timber frame flexibility, and the infills capacity to prevent excessive deformation of the frame. The two component materials are generally not tightly interconnected; thus, through the cracks in the mortar joints of the infills and by the deformations of the timber connections (which usually have gaps from the poor execution), the earthquake energy is dissipated and the house resists seismic actions. Another very important structural feature is the presence of the braces, which have a significant contribution to the overall capacity (almost doubling it, as it resulted from experimental tests [40]), although they are connected below the upper corner joint of the timber frame.”
House 2: “The seismic behavior of such a paianta house strongly depends on the presence of the diagonal braces. For this particular case study, they were missing, so most likely the capacity of the walls to lateral forces was reduced. However, as it resulted from the experimental analysis [40], the entire structural system is very ductile, allowing large displacements at the top with no visible or irreversible damages. The horizontal strips have a low bracing effect for the timber posts. Being very flexible and only attached with nails, their role is merely to keep in place the earth and straw mix. The seismic energy is dissipated through the cracks in the earthen mix, and also through the timber joints of the frame.”
House 3: “The seismic behavior of this house type would also depend on the presence of the diagonal timber braces, which are missing here as well. Nevertheless, like in the previous case study and as resulting from the experimental analysis [40], the system, even if in this case the twigs are laid on the vertical direction, is very ductile and allows quite large displacements at the top, with no visible or irreversible damage. The seismic energy is dissipated through the cracks in the infills, the twigs ‘connections to the frame and also through the timber joints of the frame.”
Comment 2: It is recommended to provide a close-up view of the crack photograph for figures 33, 34 and 35.
Answer 2: We drew the cracks with a red line, exactly as they look on the wall, for all the mentioned Figures.
Comment 3: Figure 18 and 42 are not clear. It is strongly recommended to amend those figures that only highlighting the relevant structural component or gridline.
Answer 3: We replaced the old Fig. 18 (new Fig. 29) with two lateral views (and modified the caption accordingly), hoping they are more relevant for the readers. Also, we replaced old Fig. 42 (new Fig. 40) with a zoomed version, with less dimensions.
Comment 4: It is recommended to include method used for observation of degradation. Any standards or guidelines used as references?
Answer 4: We included a sentence with the two references that we used in the field investigations:
“The field investigations were made with reference to the templates suggested in [29], [30]. “
Comment 5: The observed degradation for each case study can be included in section 5 so it is easier to be compared
Answer 5: We have included the observed degradations for each case study in a distinct section and we have reordered the manuscript sections for an improved structuring and readability. We sincerely hope this will answer the reviewer’s requirements.
Reviewer 2 Report
It is a very interesting technical paper. The authors presented an overview of the traditional building construction techniques in Romania, focusing on the structural configuration and detailing of the so-called paiantă houses. The conclusions can be used to substantiate further research aimed at selecting the most appropriate construction and strengthening techniques.
The theme of the paper is novel, the description and analysis are very clear, and the arguments are detailed. In general, it is a well written paper, despite some shortcomings.
The introduction was good, but it was not well connected with the following. In this chapter, it is better to add a paragraph to indicate the structure of the paper.
The conclusion of the paper was wordy, but the theme was not prominent, which cannot reflect the innovation of this paper. The last paragraph introduced a case or even a picture (line 700-line701), which was not suitable for the conclusion. This chapter should be revised.
Author Response
We thank the reviewer for the kind and insightful comments and suggestions, which substantially contributed to the improvement of our manuscript. We have done our best to address all the requirements. Please find below the description of the revisions that we have made. These have hopefully brought the manuscript to a level that would fully meet the reviewer’ expectations.
For an improved clarity and readability, we have also reordered and partially restructured the sections of the paper. Thus, we have created a clearer outline of the paper, as follows.
- Introduction
- Seismic performance of traditional Romanian houses
- Traditional houses: components and materials
- Types of paiantă houses
- Most common degradations found in paiantă houses
- Selected case studies
- Observed degradations for the case study houses
- Discussion
- Conclusions
We have also made a revision of the English language and of the general formatting. Small interventions were made at a few locations in the text to improve clarity and readability. All the changes were made by using the “Track Changes” feature of Microsoft Word, according to the Editors’ instructions.
We provide also a “clean” manuscript, without the “Track Changes” marks.
Comment 1: The introduction was good, but it was not well connected with the following. In this chapter, it is better to add a paragraph to indicate the structure of the paper.
Answer 1: We introduced a paragraph at the end of the Introduction Chapter:
“The paper presents the characteristic features of paianta houses, as observed during several field investigations. The field investigations were made with reference to the templates suggested in [29], [30]. Due to the variation of the construction details even among the same area or the same typology, the collaboration between both engineers and architects was necessary to draw valid conclusions, use the same terms and understand the main details that classify a house into the paianta typology. To deepen the knowledge, three case studies were selected among the most common types of paianta and structurally assessed to observe the damage and degradations. These were analyzed comparatively and integrated into a list of possible damages specific to paianta houses, providing thus a comprehensive outline of the main aspects to focus on when investigating an existing house. This list can represent a useful tool for field investigations. Based on the degradation inventory, further research will be conducted to establish solutions for the most common problems occurring in these buildings, by using contemporary scientific knowledge and techniques.”
Comment 2: The conclusion of the paper was wordy, but the theme was not prominent, which cannot reflect the innovation of this paper. The last paragraph introduced a case or even a picture (line 700-line701), which was not suitable for the conclusion. This chapter should be revised
Answer 2: The Fig. 63 and mentioning text were deleted. And we added in the Conclusions chapter the following paragraph:
“These tables can be used in future field investigations, including them in the template investigation sheets and helping specialists to check directly for the specific problems that are common for this typology.
The findings provided by this research are necessary, due to the lack of detailed information on this topic, especially from the engineering perspective. Moreover, the few available info is often misleading, not being supported by scientific methods.”
We have also revised the entire “Conclusions” section, for improved clarity and readability.
Reviewer 3 Report
The main topic of the manuscript can certainly be considered interesting in the modern mitigation of seismic risk and, mainly, in a new perspective on the conservation of cultural heritage.
However, the selected case studies are not presented correctly with regards to the building stock. In fact, no characterization of the building heritage is reported.
In particular, the authors say: “It is very difficult to determine the percentage of each type in certain areas, as it depends on the local culture, available materials and workmanship skills.”
Obviously, it should be the key step for the research and the authors should try to provide some guidance or define strategies to improve this part of the study. Otherwise, inthe reviewer's opinion, the study is not really "an engineering perspective."
In this sense, the study could be compared with other studies and methods, commonly adopted in studies on large stocks of buildings.
Globally, the manuscript can be considered an interesting reading but the study does not seem ready for publication yet. In fact, the study does not appear to be able to inform and support the next steps of ongoing research.
Author Response
We thank the reviewer for the kind and insightful comments and suggestions, which substantially contributed to the improvement of our manuscript. We have done our best to address all the requirements. Please find below the description of the revisions that we have made. These have hopefully brought the manuscript to a level that would fully meet the reviewer’ expectations.
For an improved clarity and readability, we have also reordered and partially restructured the sections of the paper. Thus, we have created a clearer outline of the paper, as follows.
- Introduction
- Seismic performance of traditional Romanian houses
- Traditional houses: components and materials
- Types of paiantă houses
- Most common degradations found in paiantă houses
- Selected case studies
- Observed degradations for the case study houses
- Discussion
- Conclusions
We have also made a revision of the English language and of the general formatting. Small interventions were made at a few locations in the text to improve clarity and readability. All the changes were made by using the “Track Changes” feature of Microsoft Word, according to the Editors’ instructions.
We provide also a “clean” manuscript, without the “Track Changes” marks.
Comment 1: However, the selected case studies are not presented correctly with regards to the building stock. In fact, no characterization of the building heritage is reported.
Answer 1: We added in the beginning of Chapter 4 this paragraph:
“Among the five identified types of paianta, the first three are the most common. All three of them have been previously investigated experimentally [40] for the behavior under combined vertical and lateral loads. Only one single wall from each type could be constructed and tested, and for this reason the variation of the results could not be properly characterized, even though some interesting conclusions could be drawn. The results of the tests showed overall a ductile behavior, with the main damage consisting of cracks in the infills. The presence of braces doubled the capacity to lateral loads. On the other side, for type 1 (timber frames with masonry infills) a sensitivity of the infills in the out-of-plane direction was noticed, explainable by the fact that, even from the beginning, the mud mortar cracks due to drying shrinkage and has no connection with the timber frame. The actual behavior in earthquakes confirmed the results of the experiments and showed that, usually, even if the infills may fall, the timber frame will be able to further support the loads. It was noticed that the timber connections, although poorly executed sometimes, can have significant deformations, however not large enough to lead to the collapse of the house [38].
To illustrate the actual condition of paianta houses located in various regions of Romania, a number of three case studies is presented in the following. Even though there are several similar houses currently inhabited and well maintained in the investigated areas, the case studies were chosen from not inhabited and not maintained houses. The degradations from various factors were more accentuated in these buildings and the structure was partly exposed due to fallen plaster, thus easier to inspect.”
Comment 2: In particular, the authors say: “It is very difficult to determine the percentage of each type in certain areas, as it depends on the local culture, available materials and workmanship skills.”
Obviously, it should be the key step for the research and the authors should try to provide some guidance or define strategies to improve this part of the study. Otherwise, in the reviewer's opinion, the study is not really "an engineering perspective."
Answer 2: We assessed the percentage of each typology in the village of Dedulesti, to provide guidance in this sense. And we changed the corresponding paragraph in the paper:
“Some of the investigated houses were abandoned, while others were well-maintained, with owners not complaining about special issues with them. Depending on the area, as noticed also during the field investigations conducted by the Order of Architects of Romania (OAR) RURAL Group in order to develop local architecture guidelines for traditional houses [39], certain types of paianta prevail, since this depends, as mentioned, on the local culture, available materials and workmanship skills. As an example, an entire village (DeduleÅŸti, Vrancea County) was considered in the investigation performed by the Technical University of Civil Engineering (UTCB). Here, most of the houses were investigated and a survey was conducted to assess the number of houses of each type that can be found in the village. Obviously, given that some of the houses with undamaged finishing the structural type could not be investigated, as owners would not allow plaster removal and/or some of the younger owners were not aware of all the constructive details of house. In this case, the type was assessed based on the similarity with other contemporary houses from the same village or ethnographic area, for which the structural type was known.”
Comment 3: In this sense, the study could be compared with other studies and methods, commonly adopted in studies on large stocks of buildings.
Answer 3: We thank the reviewer for this insightful suggestion. Our study was actually based on the large study done by OAR RURAL Group, since they have developed guides all over the country. We changed the end of the section describing the case studies and restructured the manuscript by reordering the sections in a more logical way.
Comment 4: Globally, the manuscript can be considered an interesting reading but the study does not seem ready for publication yet. In fact, the study does not appear to be able to inform and support the next steps of ongoing research.”
Answer 4: We added two paragraphs in the Conclusions:
“These tables can be used in future field investigations, including them in the template investigation sheets and helping specialists to check directly for the specific problems that are common for this typology.
The findings provided by this research are necessary, due to the lack of detailed information on this topic, especially from the engineering perspective. Moreover, the few available info is often misleading, not being supported by scientific methods.”
and
“Based on the detailed case studies and also on previous research [28], numerical models will be adapted for type 1 houses, starting from [45], while for the other types they will be further developed. Also, future experimental studies on paiantă walls will determine the variation of their response under combined lateral and vertical loads, with several specimens being built and tested. After testing, some of the walls will be strengthened using various simple methods and then tested again, to assess the efficacy of the applied retrofitting techniques.”
We also revised some other parts of the “Conclusions” sections, for improved readability and clarity.
We introduced a paragraph at the end of the Introduction Chapter:
“The paper presents the characteristic features of paianta houses, as observed during several field investigations. The field investigations were made with reference to the templates suggested in [29], [30]. Due to the variation of the construction details even among the same area or the same typology, the collaboration between both engineers and architects was necessary to draw valid conclusions, use the same terms and understand the main details that classify a house into the paianta typology. To deepen the knowledge, three case studies were selected among the most common types of paianta and structurally assessed to observe the damage and degradations. These were analyzed comparatively and integrated into a list of possible damages specific to paianta houses, providing thus a comprehensive outline of the main aspects to focus on when investigating an existing house. This list can represent a useful tool for field investigations. Based on the degradation inventory, further research will be conducted to establish solutions for the most common problems occurring in these buildings, by using contemporary scientific knowledge and techniques.”
Reviewer 4 Report
Dear authors,
The present manuscript details an in-field study about typical vernacular housing building in Romania. It conducts the engineering overall qualitative assessment of 3 paianta.
Though the reviewer is convinced that such documentation is relevant to the scientific and engineering communities, the present manuscript show too many weaknesses.
The significance of the content is limited as :
- only 3 structural qualitative assessment are conducted which lack representativity of the whole diversity of paianta
- no comparison is done with the damage evolution of concrete buildings to compare the efficiency. We don't understand if the damage typical of paianta (water issues + foundation settlement) are more or less pronounced than for other typologies.
- We don't see the structural state of well-maintained paianta to understand if they are viable for housing
- No strengthening (or improvement of the design) of the existing paianta houses is proposed
- The tables at the end of the paper show common damage to paianta, but we don't understand which one are actually expected for paianta in general.
- The authors often discuss the good performance of paianta houses against earthquakes (intro + conclusions) but never discuss it in the light of the three cases of study (diagonal elements, flexibility).
The format and quality of presentation are not adequate. The objective and contribution of the paper is not described anywhere. After reading it several times, it did not appear to me. A paper should have a clear objective and then develop a methodology to follow this objective.
I suggest to go much earlier into the topic of the paper (i.e. paianta houses).
The introduction is fuzzy and there is no clear path of "where do the authors want to go". It lacks a clear structure.
Some parts lack links each to another. The methodology should be presented after the intro.
The Figures are not well ordered (first cited, first shown) and some are even not cited. Some are also not understandable, with sometimes some text not in english.
Some references are incomplete and the state of the art shows too little references, with most of them not in english which does not help the reader to dive into the subject.
For all these reasons, I recommend the rejection of the paper.
Author Response
We thank the reviewer for the insightful comments and suggestions, which substantially contributed to the improvement of our manuscript. We respect the reviewer opinion and we have done our best to address all the requirements. Please find below the description of the revisions that we have made. These have hopefully brought the manuscript to a level that would fully meet the reviewer’ expectations.
For an improved clarity and readability, we have also reordered and partially restructured the sections of the paper. Thus, we have created a clearer outline of the paper, as follows.
- Introduction
- Seismic performance of traditional Romanian houses
- Traditional houses: components and materials
- Types of paiantă houses
- Most common degradations found in paiantă houses
- Selected case studies
- Observed degradations for the case study houses
- Discussion
- Conclusions
We have also made a revision of the English language and of the general formatting. Small interventions were made at a few locations in the text to improve clarity and readability. All the changes were made by using the “Track Changes” feature of Microsoft Word, according to the Editors’ instructions.
We provide also a “clean” manuscript, without the “Track Changes” marks.
Comment 1: only 3 structural qualitative assessment are conducted which lack representativity of the whole diversity of paianta
Answer 1: We added in the beginning of Chapter 4 this paragraph:
“Among the five identified types of paianta, the first three are the most common. All three of them have been previously investigated experimentally [40] for the behavior under combined vertical and lateral loads. Only one single wall from each type could be constructed and tested, and for this reason the variation of the results could not be properly characterized, even though some interesting conclusions could be drawn. The results of the tests showed overall a ductile behavior, with the main damage consisting of cracks in the infills. The presence of braces doubled the capacity to lateral loads. On the other side, for type 1 (timber frames with masonry infills) a sensitivity of the infills in the out-of-plane direction was noticed, explainable by the fact that, even from the beginning, the mud mortar cracks due to drying shrinkage and has no connection with the timber frame. The actual behavior in earthquakes confirmed the results of the experiments and showed that, usually, even if the infills may fall, the timber frame will be able to further support the loads. It was noticed that the timber connections, although poorly executed sometimes, can have significant deformations, however not large enough to lead to the collapse of the house [38].”
Comment 2: no comparison is done with the damage evolution of concrete buildings to compare the efficiency. We don't understand if the damage typical of paianta (water issues + foundation settlement) are more or less pronounced than for other typologies.
Answer 2: We have added a paragraph at the end of Chapter 8:
“When compared to the popular typology of nowadays, reinforced concrete frames with masonry infills, the paianta houses may need more maintenance during time. Because the concrete buildings appeared more recent than the traditional ones, most of them have concrete foundations, which generally prevent differential settlement, which affects many paianta houses. However, even for concrete houses, if they are not inhabited and well maintained, in time they also develop dampness inside. And from the seismic point of view, sometimes, like in the case of Haiti earthquake in 2010, the traditional houses showed a more resilient behavior and when the infills collapse the timber frame still stands and supports the roof. Such a behavior does not usually kill people, unlike the poorly constructed concrete houses that produced many casualties.”
Comment 3: - We don't see the structural state of well-maintained paianta to understand if they are viable for housing
Answer 3: In Chapter 4, a sentence and a photo were added:
“An example of a maintained traditional house, which is also inhabited, is shown in Fig. 20, but the structural type of paianta is not visible.”
And also the following figure:
Fig. 20. Inhabited and well maintained traditional house in Voinesti village, Dambovita County (@ architect Cornelia Zaharia)
Comment 4: - No strengthening (or improvement of the design) of the existing paianta houses is proposed
Answer 4: This is the topic of our next paper which is now being written, and it will start from the 3 case studies presented in this paper. The strengthening topic is quite wide and if we approach it in this paper it will become too long.
However, we added the following paragraphs at the end of Chapter 6:
“House 1 has structural problems already due to the advanced biological decay of the mudsill, and it needs replacement of the decayed timber elements and dismantling of the existing brick masonry, which is not difficult due to the weak mud mortar. Then, the infills will be restored using the dismantled bricks. This work needs to consider additional support for the roof framework. Besides this, foundation underpinning is necessary to stop the differential settlement producing cracks in the walls.
Among the studied cases, the second house appears to be less affected by infiltrations, although more biological tests are required to determine if mold and fungus is already present. With minimum interventions to replace the degraded roof cover on the backside, filling the cracks from the backside wall with earth and straw mix, replastering it and ensuring constant heating during winter, the house can easily become inhabitable again.
The repairing process proposed for House 3 aims to restore the house to its initial structural state by removing the inappropriate additions and replacing the deteriorated parts. According to [20], the restauration works will consist of giving the structural elements their original strength, within the prevention and conservation objective. The restoration process allows structural additions, such as timber braces, when proved necessary by the structural evaluation done by an expert. Some of the most common works will be replacing the roof cover, repairing the affected post and the façade in the corresponding area, removing the concrete socle, creating a sidewalk and adding gutters continued by downspouts to remove the water near the bottom part of the house.”
Comment 5: The tables at the end of the paper show common damage to paianta, but we don't understand which one are actually expected for paianta in general.
Answer 5: At the end of the former Section 5 we modified the concluding paragraph and restructured it in a “Discussion” section, introducing the following text:
“The three case studies presented in this paper have shown that local mold and dampness are one of the most common degradation-causing factors. These appear due to water infiltrations, mainly caused by poor maintenance. All the considered houses are not inhabited; thus, they are not heated during the cold seasons, which also favors excessive indoor humidity. It should be noted, however, that poor maintenance generally leads to similar effects, no matter the building age or type.”
Comment 6: The authors often discuss the good performance of paianta houses against earthquakes (intro + conclusions) but never discuss it in the light of the three cases of study (diagonal elements, flexibility).
Answer 6: We added in the beginning of Chapter 6 (according to the new arrangement of the chapters) this paragraph:
“Among the five identified types of paianta, the first three are the most common. All three of them have been previously investigated experimentally [40] for the behavior under combined vertical and lateral loads. Only one single wall from each type could be constructed and tested, and for this reason the variation of the results could not be properly characterized, even though some interesting conclusions could be drawn. The results of the tests showed overall a ductile behavior, with the main damage consisting of cracks in the infills. The presence of braces doubled the capacity to lateral loads. On the other side, for type 1 (timber frames with masonry infills) a sensitivity of the infills in the out-of-plane direction was noticed, explainable by the fact that, even from the beginning, the mud mortar cracks due to drying shrinkage and has no connection with the timber frame. The actual behavior in earthquakes confirmed the results of the experiments and showed that, usually, even if the infills may fall, the timber frame will be able to further support the loads. It was noticed that the timber connections, although poorly executed sometimes, can have significant deformations, however not large enough to lead to the collapse of the house [38].”
Then, we added a paragraph for each case study (at the end) explaining the seismic performance of the buildings:
“House 1: “The configuration of the structure showed a ductile behavior in previous seismic events. No typical seismic damage was observed, but only degradations due to water infiltrations and lack of maintenance. Even with the timber skeleton partially degraded and the brick masonry infill also partially damaged, the house is still able to withstand seismic motions. The key feature of this type of paianta is the timber frame flexibility, and the infills capacity to prevent excessive deformation of the frame. The two component materials are generally not tightly interconnected; thus, through the cracks in the mortar joints of the infills and by the deformations of the timber connections (which usually have gaps from the poor execution), the earthquake energy is dissipated and the house resists seismic actions. Another very important structural feature is the presence of the braces, which have a significant contribution to the overall capacity (almost doubling it, as it resulted from experimental tests [40]), although they are connected below the upper corner joint of the timber frame.”
House 2: “The seismic behavior of such a paianta house strongly depends on the presence of the diagonal braces. For this particular case study, they were missing, so most likely the capacity of the walls to lateral forces was reduced. However, as it resulted from the experimental analysis [40], the entire structural system is very ductile, allowing large displacements at the top with no visible or irreversible damages. The horizontal strips have a low bracing effect for the timber posts. Being very flexible and only attached with nails, their role is merely to keep in place the earth and straw mix. The seismic energy is dissipated through the cracks in the earthen mix, and also through the timber joints of the frame.”
House 3: “The seismic behavior of this house type would also depend on the presence of the diagonal timber braces, which are missing here as well. Nevertheless, like in the previous case study and as resulting from the experimental analysis [40], the system, even if in this case the twigs are laid on the vertical direction, is very ductile and allows quite large displacements at the top, with no visible or irreversible damage. The seismic energy is dissipated through the cracks in the infills, the twigs ‘connections to the frame and also through the timber joints of the frame.”
Comment 7: The format and quality of presentation are not adequate. The objective and contribution of the paper is not described anywhere. After reading it several times, it did not appear to me. A paper should have a clear objective and then develop a methodology to follow this objective.
Answer 7: We added two paragraphs in the Conclusions:
“These tables can be used in future field investigations, including them in the template investigation sheets and helping specialists to check directly for the specific problems that are common for this typology.
The findings provided by this research are necessary, due to the lack of detailed information on this topic, especially from the engineering perspective. Moreover, the few available info is often misleading, not being supported by scientific methods.”
and
“Based on the detailed case studies and also on previous research [28], numerical models will be adapted for type 1 houses, starting from [45], while for the other types they will be further developed. Also, future experimental studies on paiantă walls will determine the variation of their response under combined lateral and vertical loads, with several specimens being built and tested. After testing, some of the walls will be strengthened using various simple methods and then tested again, to assess the efficacy of the applied retrofitting techniques.”
We introduced a paragraph at the end of the Introduction Chapter:
“The paper presents the characteristic features of paianta houses, as observed during several field investigations. The field investigations were made with reference to the templates suggested in [29], [30]. Due to the variation of the construction details even among the same area or the same typology, the collaboration between both engineers and architects was necessary to draw valid conclusions, use the same terms and understand the main details that classify a house into the paianta typology. To deepen the knowledge, three case studies were selected among the most common types of paianta and structurally assessed to observe the damage and degradations. These were analyzed comparatively and integrated into a list of possible damages specific to paianta houses, providing thus a comprehensive outline of the main aspects to focus on when investigating an existing house. This list can represent a useful tool for field investigations. Based on the degradation inventory, further research will be conducted to establish solutions for the most common problems occurring in these buildings, by using contemporary scientific knowledge and techniques.”
Comment 8: I suggest to go much earlier into the topic of the paper (i.e. paianta houses).
Answer 8: We rearranged the introduction, breaking it in 2 sections: “Introduction” and “Seismic performance of traditional Romanian houses”, for improved structuring and for enhanced clarity and readability.
Comment 9: The introduction is fuzzy and there is no clear path of "where do the authors want to go". It lacks a clear structure.
Answer 9: Please see answer 7 above. We thank the reviewer for this insightful suggestion. We reorganized the introduction and created a second section dealing with the seismic performance of traditional Romanian houses.
Comment 10: Some parts lack links each to another. The methodology should be presented after the intro.
Answer 10: We added a paragraph at the end of the Introduction to make the purpose of the paper clearer.
“The paper presents the characteristic features of paianta houses, as observed during several field investigations. The field investigations were made with reference to the templates suggested in [29], [30]. Due to the variation of the construction details even among the same area or the same typology, the collaboration between both engineers and architects was necessary to draw valid conclusions, use the same terms and understand the main details that classify a house into the paianta typology. To deepen the knowledge, three case studies were selected among the most common types of paianta and structurally assessed to observe the damage and degradations. These were analyzed comparatively and integrated into a list of possible damages specific to paianta houses, providing thus a comprehensive outline of the main aspects to focus on when investigating an existing house. This list can represent a useful tool for field investigations. Based on the degradation inventory, further research will be conducted to establish solutions for the most common problems occurring in these buildings, by using contemporary scientific knowledge and techniques.”
Comment 11: The Figures are not well ordered (first cited, first shown) and some are even not cited. Some are also not understandable, with sometimes some text not in english.
Answer 11: We rearranged the Figures in the right order and replaced the ones that were not understandable. We provided captions with translated text for figures in which text was not in English. We revised the citations.
Comment 12: Some references are incomplete and the state of the art shows too little references, with most of them not in english which does not help the reader to dive into the subject.
Answer 12: There are not many publications in English about Romanian paianta houses and most of the scientific literature is published in Romanian. We translated the titles of the references that we used, even if they are in Romanian.
We added in the “Introduction” section the following paragraph:
“During the past decade, several studies were conducted on vernacular practices in the field, with a special focus on the experimental assessment of their behavior. These studies comprised the Haitian kay peyi, discussed in [4], the casa baraccata in Calabria, Italy in [5] or the dhajji dewari in the mountainous areas of South Asia in [6], [7]. A reference guidebook on dhajji dewari. aimed for the use of technicians and artisans, was prepared by in [8], based on results from extensive studies including shake table tests and synthesizing the expertise gained from the over 120,000 rural houses that were rebuilt after the 2005 Kashmir earthquake using this construction technique. In [9] and [10] experimental research was performed on the traditional Chuan Dou timber structures; the Peruvian quincha buildings were studied in[11]; [12], [13] and [14] among others, focused on Portuguese pombalino buildings, while [15] shows the experimental research conducted on the Turkish himis buildings. Similar types of structural systems were used for centuries, under various names, also in other parts of the globe, as pointed out by Gülkan and Langenbach in [2]. A thorough field investigation on traditional houses in Greece was described in [16].”
The text from the Figures that were not translated was added in the caption as the Legend, and text boxes with translation were added over the Figure 37.
Reviewer 5 Report
The paper is very interesting. Maybe the author could intrduce readers with a map indicating which are the study houses, and given an idea about the seismic maps and the environmental data, for instance: rain, temperature and s on.

Author Response
We thank the reviewer for the kind and insightful comments and suggestions, which substantially contributed to the improvement of our manuscript. We have done our best to address all the requirements. Please find below the description of the revisions that we have made. These have hopefully brought the manuscript to a level that would fully meet the reviewer’ expectations.
For an improved clarity and readability, we have also reordered and partially restructured the sections of the paper. Thus, we have created a clearer outline of the paper, as follows.
- Introduction
- Seismic performance of traditional Romanian houses
- Traditional houses: components and materials
- Types of paiantă houses
- Most common degradations found in paiantă houses
- Selected case studies
- Observed degradations for the case study houses
- Discussion
- Conclusions
We have also made a revision of the English language and of the general formatting. Small interventions were made at a few locations in the text to improve clarity and readability. All the changes were made by using the “Track Changes” feature of Microsoft Word, according to the Editors’ instructions.
We provide also a “clean” manuscript, without the “Track Changes” marks.
Comment 1: Maybe the author could intrduce readers with a map indicating which are the study houses, and given an idea about the seismic maps and the environmental data, for instance: rain, temperature and s on.
Answer 1: The required map and data were introduced.
The following paragraphs were introduced in the section dedicated to the case study descriptions:
“The map in Fig. 23 shows the locations of the case study houses. All of them are situated in rural or peri-urban areas of the extra-Carpathian part of Romania. The first two are typical for the plain and hilly zones of Muntenia, in the southern part of the country, while the third one is located near the port town of Sulina, in the Danube Delta.
(for the Figure 23, please check the revised manuscript)
Fig. 23. Locations of houses considered for the case studies, with local topography shown on the map
The seismicity of the above areas is significant, as they are quite close to the Vrancea seimogenic source, located at the Carpathian arc bend. Table 7 shows the values of the peak ground acceleration, ag, and of the characteristic (corner) period, TC, according to the Romanian seismic design code, P100-1/2013 [35].
Table 7. Design values of peak ground acceleration, ag, and characteristic (corner) period, TC, according to the Romanian seismic design code, P100-1/2013, for the locations of case study houses
|
Case Study |
Peak ground acceleration, ag [g] |
Characteristic (corner) period, TC [s] |
|
1. Tufani Village |
0.35 |
1.6 |
|
2. Ion Roată Village |
0.35 |
1.6 |
|
3. Sulina Town |
0.20 |
0.7 |
As it can be noticed from Table 7, all case study houses and, in particular, the first two, are located in areas characterized by rather high ag values. As regards the TC value of 1.6 s for the first two sites, these were introduced in the seismic code to account for the narrow-band frequency content, concentrated at short frequencies/long periods, observed during the past strong earthquakes in Bucharest and in large areas of the Romanian Plain. Given the short natural periods of the analyzed structural typologies, this could be also considered as one of the explanations of the absence of severe seismic damage and collapse in such buildings, as already shown in Section 2 of the paper. For the third case study, even though a design value TC=0.7 s is specified in the seismic code for the entire south-eastern area of the country, local site effects related to deltaic alluvium could have led in Sulina to the shift of spectral peaks towards longer periods, thus to less aggressive effects of strong earthquakes for this short-period house typology.
A detailed description of the seismicity of Romania, including the ag zoning map according to the P100-1/2013 code, can be found in [41].
Regarding the climate for the locations of the studied houses, this is characteristic to that of the Romanian Plain (Lower Danube Plain) for the first two case studies and to the Danube Delta for the third. The climate of the Romanian Plain is temperate-continental, with the average annual temperature ranging from 11-11.5°C in the south to 10.5° in the north [42]. The mean rainfall is about 500 mm / year [42]. The climate of Sulina, in the Danube Delta, is continental, with strong influences from the vicinity of the Black Sea. The climate is characterized by an average annual temperature of 11°C, with the highest average temperature of 22°C in July and the lowest average temperature of -1°C in January [43]. The mean rainfall is of about 350 mm/year. According to the Köppen climate classification, the first two case study houses are located in Dfb areas (wet temperate continental), while the third is located in a Dfa (wet warm continental) area [44].“
References were added in the bibliography, to document these paragraphs.
Round 2
Reviewer 4 Report
Thanks to a complete restructuration, the paper is now ready to be published.
Though it is a very interesting topic, the reviewer does think that the paper could have included more pictures of more case studies instead of detailing precisely only three of them. Yet, this is only a personal opinion.